# DSTGN: Decoupled SpatioTemporal Graph Network for Multi-scenario Dynamics Learning

## Abstract

Machine Learning (ML) methods have played a pivotal role in a wide range of dynamics tasks, such as physical simulation, multi-modal interaction, and real-time prediction. Among them, Graph Neural Networks (GNNs) have emerged as effective surrogates for numerical solvers, thanks to their ability to model pairwise interactions between nodes and their neighboring edges. However, the inherent tendency of GNNs to perform local aggregation limits their expressiveness in capturing intricate global interactions. To address this limitation, we propose a general framework, **D**ecoupled **S**patio**T**emporal **G**raph **N**etwork (DSTGN), to enhance prediction performance across various downstream tasks. DSTGN enhances spatial feature representations by leveraging the channel-wise spatial interaction information in the latent spaces. In parallel, it introduces a learnable temporal integration mechanism that decouples inter-step dynamics at each latent layer, effectively mitigating error accumulation during autoregressive inference. Extensive experiments on multiple types of benchmarks demonstrate that DSTGN consistently outperforms existing baselines in both accuracy and generalization, across regular and irregular domains, as well as static and dynamic meshes.

## 1 Introduction

Dynamics learning commonly models the complex coupling between spatial locations and their temporal evolution, and is widely encountered in numerous intricate tasks such as climate modeling (Scher, 2018; Schultz et al., 2021; Nguyen et al., 2024), fluid dynamics simulation (Wang & Sun, 2018; Zheng et al., 2020; Zingaro et al., 2024), and short-term earthquake prediction (Zhang et al., 2019; Liu et al., 2024c). Beyond traditional numerical solvers, Deep Learning (DL) methods (Grover et al., 2015; Bronstein et al., 2017a), particularly Graph Neural Networks (GNNs) (Bruna et al., 2014; Li et al., 2015), have exhibited strong capabilities in solving diverse dynamics tasks (Hamilton et al., 2017; Corso et al., 2024), owing to their significant nonlinear expressiveness on complex domains.

As a key paradigm of geometric deep learning (Bronstein et al., 2017b), GNNs typically employ permutation-invariant operations (e.g., *mean, max, min*) (Maron et al., 2018; Xu et al., 2019) during message aggregation to preserve group invariance (Cohen et al., 2019). However, unlike CNNs, these predominant aggregation strategies often underutilize rich neighboring information, leading to common issues such as over-smoothing and over-squashing (Noutahi et al., 2020; Velingker et al., 2024). While many works (Corso et al., 2020; Zeng et al., 2023; Chen et al., 2024) attempt to mitigate these problems, their improvements are often marginal and accompanied by substantial computational overhead. For instance, the Principal Neighbourhood Aggregation (PNA) model (Corso et al., 2020) introduces a unique degree-scaler module to amplify or dampen node signals, yet this comes at a high cost in both time and memory.

Meanwhile, although prior research (Dehmamy et al., 2019; Corso et al., 2020; Zeng et al., 2023) have proved the importance of preserving sufficient information within local neighborhoods mathematically and empirically, most existing methods (Corso et al., 2020; He et al., 2023; Liao et al., 2024) focus only on exploiting basic latent features, rather than capturing deeper expressive representations from them. For example, while multi-scale patterns have proven effective on regular grids (Kondor & Pan, 2016; Liu et al., 2022), their extension to irregular graphs often yields limited performance improvement (Yang et al., 2022; Fortunato et al., 2022). Therefore, recent studies (Bodnar et al., 2021; Zhang et al., 2023) have made numerous attempts (e.g., the use of Weisfeiler-Lehman (WL)

Figure 1: Schematic diagram of DSTGN, which consists of the encoder, the Spatial block (S-block), the Temporal block (T-block), and the decoder. The S-block includes the S1 Processor and the S2 Processor while the T-block contains the T1 Processor. Their full architecture diagrams refer to Appendix Figure S1. Here, $\mathbf{u}_{t_k}$ denotes the input at $t_k$; $\hat{\mathbf{u}}_{t_{k+\eta}}$ the predicted state at $t_{k+\eta}$; $\mathbf{h}^l, \mathbf{h}^{l+1,*}, \mathbf{h}^{l+1}, \bar{\mathbf{h}}^{l+1}$ latent node features; $\mathbf{e}^l$ latent edge features; $\phi_v^l, \phi_e^l, \phi_{evo}^l$ differentiable functions; $\phi_s^l, \phi_t^l$ the S-block and the T-block at layer $l$.

mechanisms on countable feature spaces (Truong & Chin, 2024))); however, the resulting multi-level interactions still fail to fully capture highly coupled and implicit information in complex systems.

To address these limitations, we propose **D**ecoupled **S**patio**T**emporal **G**raph **N**etwork (DSTGN), as shown in Figure 1, to enhance prediction performance on both regular and irregular domains as well as static and dynamic meshes. We adopt the Encoder-Processor-Decoder architecture (Pfaff et al., 2021; Brandstetter et al., 2022; Mi et al., 2025) as the backbone of our framework. DSTGN decouples the entangled spatiotemporal problem into space and time for separate processing. For the spatial domain, we introduce a latent spatial learning module inspired by the node-wise message-passing mechanism in physical spaces to establish a pathway for channel-wise interactions in latent spaces, enriching spatial learning. For the temporal domain, we introduce a learnable temporal integration mechanism, which generalizes the predictor–corrector scheme into the neural network framework, to guide layer-wise feature evolution. Contributions are summarized as follows:

- We propose a new spatial block that includes a channel-wise feature interaction learning module in the latent space as a supplement to standard message-passing-based GNNs, which, as a result, strengthens DSTGN's spatial learning capacity.
- We regard the layer-wise feature update as a temporal marching process and thus establish a learnable predictor-corrector time integration scheme to improve DSTGN's temporal modeling capacity, thereby reducing the autoregressive error accumulation.
- Extensive experiments across various scenarios (e.g., 2D and 3D spaces, varying temporal intervals, regular and irregular domains, static and dynamic meshes) demonstrate superior performance and strong generalization capability of DSTGN.

## 2 RELATED WORK

**Numerical methods:** As a classical problem, dynamics modeling has long attracted significant research interest. Traditional numerical approaches (e.g., finite difference (Anderson & Wendt, 1995), finite volume (Edwards & Rogers, 1998; Eymard et al., 2000), finite element (Zienkiewicz et al., 2005), and spectral methods (Karniadakis & Sherwin, 2005)) are increasingly hindered by their substantial computational costs. The rigorous requirement of stability, consistency, and convergence criteria limits their scalability in practice.

**Data-driven methods:** Over the past few decades, neural network-based methods have shown considerable potential to overcome challenges of traditional methods. Conventional grid-based techniques (Sundermeyer et al., 2012; Koutnik et al., 2014) have been widely used to model complex dynamics systems (Albawi et al., 2017; Tolstikhin et al., 2021; Liu et al., 2024b). Meanwhile, GNN-based and Transformer-based methods (Liu et al., 2020; Gao et al., 2022) have been developed to effectively handle complex information on irregular meshes. Message Passing Neural Networks (MPNNs) (Gilmer et al., 2017b; Gao et al., 2021; Horie & Mitsume, 2024) provide a general modeling mechanism, with MeshGraphNets (Pfaff et al., 2021) and MP-PDE solver (Brandstetter et al., 2022) being prominent examples. Moreover, the "message" also passes on simplicial complexes

(SCs) (Bodnar et al., 2021). As another paradigm, the Transformer model (Vaswani et al., 2017) and its variants (Cao, 2021b; Nguyen et al., 2022; Ovadia et al., 2023), have also served as powerful surrogate models for dynamics evolution. Concretely, a non-symmetric kernel function (Hao et al., 2023) is constructed via the self attention mechanism (Vaswani et al., 2017; Lin et al., 2024) for many downstream tasks, such as approximating the integral form of Green's function for physical simulation (JANNY et al., 2023; Hoover et al., 2024). While successful in spatial modeling (e.g., leveraging the predefined "U" architecture (Su et al., 2024) or the preprocessed patch (Ovadia et al., 2023) to capture complex structures and patterns), their temporal applications remain limited (e.g., multi-input temporal attention) and lack strong theoretical support.

**Neural operators:** Neural operators aim to learn mappings between the infinite-dimensional function spaces. For example, DeepONet (Lu et al., 2021; Diab & Al Kobaisi, 2024) seeks to achieve this goal based on the universal operator approximation theorem (Chen & Chen, 1995). Moreover, Graph Neural Operator (GNO) (Li et al., 2020b; Cheng & Peng, 2024) and Fourier Neural Operator (FNO) (Li et al., 2020a; Wen et al., 2022; Tran et al., 2023; Liu et al., 2024a) approximate the solutions by learning spectral transforms in function space. In addition, wavelet bases (Su et al., 2024) have been employed to learn approximations between infinite-dimensional function spaces under the Green's function theory (Gupta et al., 2021; Cao, 2021b; Kuibarov et al., 2024). Recently, many ensemble models (Cao et al., 2024; Raonic et al., 2024; Li et al., 2024b) have been developed in an effort to better solve various downstream problems.

## 3 METHODOLOGY

**Problem statement:** Generally, the typical dynamics problem (e.g., fluid simulation) is to fit a multi-variables function $\mathcal{F}(\cdot)$ which involves itself, one or more of its partial derivatives on the $m$-dimensional space (e.g., $\mathbf{x} \in \mathbb{R}^m$) and 1-dimensional time (e.g., $t \in \mathbb{R}^1$), described as: $\partial \mathbf{u}/\partial t = \mathcal{F}(\theta, t, \mathbf{x}, \partial \mathbf{u}/\partial_x, \dots)$, where $\mathbf{u}(\mathbf{x}, t) \in \mathbb{R}^d$ is the spatiotemporal variable and $\theta$ is the related parameters in the function $\mathcal{F}(\cdot)$. Across regular and irregular domains as well as static and dynamic meshes, our research task is that: ***Given a random initial condition (IC), we perform autoregressive multi-step (e.g., 1,000 time steps) inferences without any prior knowledge***.

### 3.1 DECOUPLED SPATIOTEMPORAL GRAPH LEARNING (DSTGN)

To address the above mentioned spatiotemporal problem, we propose a general neural network, which we refer to as DSTGN, with two key objectives: (1) exploring the spatial connectivity and the temporal transmission as much as possible and (2) achieving a greater generalization without retraining under varying initial/boundary conditions and coefficients. As shown in Figure 1, our network consists of the Encoder, the Processor that includes the Spatial block (S-block) and the Temporal block (T-block), and the Decoder to learn the complex dynamics. Here, the Encoder and the Decoder model the mappings between latent features and physical variables, defined as follows. The S-block is designed for leveraging the rich spatial information and the T-block is to guide the temporal transmission of features. A detailed procedure of DSTGN is provided in Appendix Algorithm 1. For simplicity, routine operations such as normalization and activation functions are omitted. The source code and data can be found at Github (posted after the peer-review process).

**Encoder:** We design the encoder block with differentiable functions (e.g., MLPs) to map the low-dimensional variables to high-dimensional latent features. For clarity, the initial node (0-simplex) feature $\mathbf{h}_i^0 \in \mathbb{R}^{1 \times d_0}$ is configured by the node variables $\mathbf{u}_i$, one-hot feature of node type $\kappa_i$, and their position information $\mathbf{x}_i$. The initial edge (1-simplex) feature $\mathbf{e}_{ij}^0 \in \mathbb{R}^{1 \times d_0}$ is generated by the relative position vector and its norm. The corresponding forms are described as:

$$\mathbf{h}_i^0 = \phi_v^{en}(\mathbf{u}_i \parallel \mathbf{x}_i \parallel \kappa_i \parallel \dots), \quad \mathbf{e}_{ij}^0 = \phi_e^{en}((\mathbf{x}_j - \mathbf{x}_i) \parallel d_{ij} \parallel \dots), \tag{1}$$

where functions $\phi_v^{en}(\cdot)$ and $\phi_e^{en}(\cdot)$ are applied to learn the latent features of node and edge; $(\mathbf{x}_j - \mathbf{x}_i)$ a relative position vector between the nodes $i$ and $j$; $d_{ij}$ the relative physical distance; $(\cdot \parallel \cdot)$ the concatenation operation; $d_0$ the dimension of initial latent features.

**Decoder:** We design the decoder block to map latent features back to physical variables on graphs. The new states $\mathbf{u}_{t_{k+1}}$ is obtained by the incremental learning scheme, described as:

$$\hat{\mathbf{u}}_{t_{k+1}} = \phi_v^{de}(\mathbf{h}^L) + \mathbf{u}_{t_k}, \tag{2}$$

where $\phi_v^{de}(\cdot)$ is a differentiable function (e.g., MLPs) and $L$ is the total number of processor layers.

## 3.2 SPATIAL BLOCK (S-BLOCK)

Consider a discretized graph $G = (V, E, A)$, where $V \in \mathbb{R}^{|V| \times d_v}$ is the set of nodes, $E \in \mathbb{R}^{|E| \times d_e}$ is the set of edges, $A \in \mathbb{R}^{|V| \times |V|}$ is the adjacency matrix, the $|V|$ is the number of nodes, $|E|$ is the number of edges, $d_v$ and $d_e$ are the dimension of node and edge features. This graph structure (initially defined by a set of nodes and edges) can be enriched by introducing virtual topological elements (e.g., adjacent triangles), as shown in Figure 2**a**. This transformation allows the network to capture richer local relationships and recover finer continuous spatial details. To mathematically formulate this transition from a discrete graph to a continuous space, we draw inspiration from finite-element concepts (Edwards & Rogers, 1998). Let $\mathcal{F}(\cdot)$ represent an attribute function over a continuous region $\Omega$. We approximate the integral of $\mathcal{F}$ as:

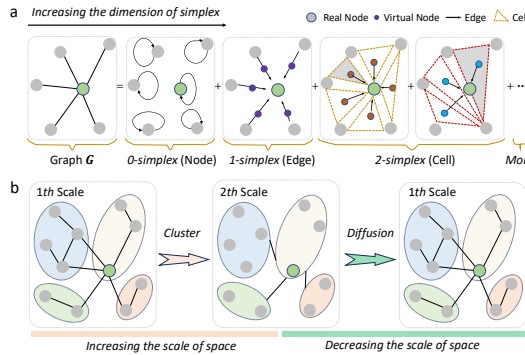

Figure 2: Diagram of decoupling graphs. **a**, the case of the same scale. **b**, the case of varying scales. More details refer to Appendix Section B.

$$\int_{\Omega} \mathcal{F}(\cdot) \, d\gamma \approx \sum_{v \in V} \mathcal{F}(\cdot) \Delta\gamma_v + \sum_{e \in E} \mathcal{F}(\cdot) \Delta\gamma_e + \sum_{t \in T} \mathcal{F}(\cdot) \Delta\gamma_t + \cdots, \tag{3}$$

where $T$ denotes the set of virtual topological elements (e.g., triangles), and the ellipsis ($\cdots$) indicates potential higher-order terms. These virtual elements can be derived directly from the preprocessed graph structure, allowing each node to integrate diverse local contextual information. This approach leverages fundamental topological principles and aligns with the insight that the space continuity is essential in graph learning (Corso et al., 2020; Nikolentzos & Skianis, 2025).

However, incorporating more higher-order terms leads to rapidly escalating computational complexity as the simplex dimension increases (Bodnar et al., 2021; Mi & Sun, 2024). Therefore, we retain only the first two terms in Eq. 3, respectively corresponding to aggregated node- and edge-level information, which suffice to capture the necessary spatio-structural patterns. These representations are then fed into the processor for subsequent dynamics prediction.

**S1 Processor:** We employ the message-passing mechanism (Pfaff et al., 2021) to construct the spatial processor block (S1) for iteratively processing the features from the upstream block. For concreteness, each node can learn information from itself and its neighbor edges, described as follows:

$$\check{\mathbf{h}}_i^{l+1} = \phi_v^l \left( \mathbf{h}_i^l \, \| \, \sum_{j \in \mathcal{N}_i} \mathbf{e}_{ij}^{l+1} \right) \quad \text{with} \quad \mathbf{e}_{ij}^{l+1} = \phi_e^l \left( \mathbf{h}_i^l \, \| \, \mathbf{h}_j^l \, \| \, \mathbf{e}_{ij}^l \right), \tag{4}$$

where $\mathcal{N}_i$ represents the set of neighboring indices of node $i$; $\phi_v^l(\cdot)$ and $\phi_e^l(\cdot)$ are differentiable functions (e.g., MLPs). The updated node features $\check{\mathbf{h}}_i^{l+1}$ are thus derived directly from the 0-simplex (node) and 1-simplex (edge) features, without incorporating other higher-order simplices.

### 3.2.1 SPATIAL LEARNING IN LATENT SPACES

However, the feature degradation issue in the message-passing mechanism may limit spatial learning, necessitating a new module to relieve it. Drawing inspiration from Figure 2**b**, which describes the clustering and diffusion of node information in physical spaces, we posit that high-dimensional latent features may exhibit analogous relational patterns. We thus generalize this phenomenon from the physical space to the latent space and accordingly propose a spatial learning block (S2) operating on latent representations.

**S2 Processor:** Consider a node in the layer $l$ with $d_l$-dimensional feature values. We (1) identify a compressible feature space by grouping these features into $k_l$ classes (channel cluster rather than node cluster) and record the transformation matrix $\mathbf{M} \in \mathbb{R}^{k_l \times d_l}$ and (2) learn its corresponding "evolutive" transformation matrix $\phi_{evo}^l(\mathbf{M}) \in \mathbb{R}^{k_l \times d_l}$ in high-dimensional latent spaces with a low complexity, and (3) finally map the features back from the compressed space to the original space. This process is formulated as:

$$\bar{\mathbf{h}}^{l+1} = (\mathbf{h}^l \mathbf{W}^l) \left( \phi_{evo}^l(\mathbf{M}) \right) \quad \text{with} \quad \mathbf{M} = (\mathbf{h}^l \mathbf{W}^l)^T (\mathbf{h}^l), \tag{5}$$

where $\mathbf{W}^l \in \mathbb{R}^{d_l \times k_l}$ is the transformation function and $\phi_{evo}^l$ is a differentiable block (e.g., MLPs, Transformer, Mixer) to facilitate a deeper understanding of the underlying dynamics.

In summary, each S-block $\phi_s^l$ integrates the S1 and S2 Processors to transform input features $(\mathbf{h}^l, \mathbf{e}^l)$ into updated node features $\mathbf{h}^{l+1,*}$. This mapping is formally defined as $\phi_s^l : (\mathbf{h}^l, \mathbf{e}^l) \mapsto \mathbf{h}^{l+1,*}$, where the output is obtained by summing the latent features from both processors:

$$\mathbf{h}^{l+1,*} = \check{\mathbf{h}}^{l+1} + \bar{\mathbf{h}}^{l+1}. \tag{6}$$

Here, $\check{\mathbf{h}}^{l+1}$ and $\bar{\mathbf{h}}^{l+1}$ denote the outputs of the S1 and S2 Processors, respectively. This design enables the model to effectively leverage the latent spatial information to enhance the generalization capability. See Table 2 for support details of its ablation study.

### 3.3 TEMPORAL BLOCK (T-BLOCK)

As shown in Figure 3, the transmission sequence in a single-layer network with a corresponding function $\phi^l$ can be described via the autoregressive update form over time:

$$\mathbf{h}^{l+1} = \mathbf{h}^l + \phi^l(\mathbf{h}^l), \tag{7}$$

where $\mathbf{h}^l$ and $\mathbf{h}^{l+1}$ represent the input and output latent features, respectively. This formulation corresponds to a standard explicit time integration scheme, which directly computes the next state based on the current state. While this makes training straightforward, it is prone to instability. In contrast, implicit methods operating on both current and future states offer improved stability, but at a higher computational cost (Bathe & Baig, 2005), precluding their use in the present network design. The limitation of explicit schemes, coupled with the prohibitive expense of implicit methods, motivates our design of a new temporal mechanism that enhances stability while maintaining efficiency.

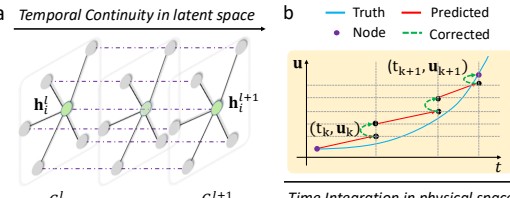

Figure 3: Diagram of temporal transmission. **a**, intermediate states exist between layers. $\mathbf{h}_i^l$ in $G^l$ and $\mathbf{h}_i^{l+1}$ in $G^{l+1}$ are two nodes at the same position. **b**, the case of time integration strategy. The red line is the forward prediction based on the last point (e.g., $(t_k, \mathbf{u}_k)$), the green dot line represents a corrected function based on the last point and the predicted point, the blue line is the real function.

#### 3.3.1 TEMPORAL STRATEGY

To achieve this purpose, we conceptualize the layer-wise feature update as a temporal marching process and introduce a unique learnable temporal block (T-block) inspired by the numerical algorithm "predictor-corrector" (Gragg & Stetter, 1964) to guide the temporal transmission. Given on the well-studied theoretical basis, our learnable temporal strategy effectively improves the model's temporal modeling capacity and reduces the autoregressive error accumulation. More details of the numerical time integration refer to Appendix Section C. Our proposed strategy is formally defined as follows:

**Definition 3.1** (**A learnable temporal strategy**). Suppose that $\phi_s^l$ and $\phi_t^l$ are differentiable functions, a more stable network transmission sequence within a single layer is described as follows:

$$\textbf{Predictor: } \mathbf{h}^{l+1,*} = \phi_s^l(\mathbf{h}^l, \mathbf{e}^l), \quad \textbf{Corrector: } \mathbf{h}^{l+1} = \phi_t^l(\mathbf{h}^{l+1,*}, \mathbf{h}^l), \tag{8}$$

where $\mathbf{h}^l \in \mathbb{R}^{N \times d_l}$, $\mathbf{e}^l \in \mathbb{R}^{|E| \times d_l}$, $\mathbf{h}^{l+1} \in \mathbb{R}^{N \times d_l}$, and $\mathbf{h}^{l+1,*} \in \mathbb{R}^{N \times d_l}$ are latent features. Here, the **Predictor** is to predict the intermediate features $\mathbf{h}^{l+1,*}$ with the S-block $\phi_s^l$ and the **Corrector** is to correct the features $\mathbf{h}^{l+1,*}$ via the T-block $\phi_t^l$ to produce the output features $\mathbf{h}^{l+1}$.

*Remark* 3.2. Our learnable strategy offers improved stability over standalone explicit methods while avoiding the high cost of fully implicit methods. The local truncation error analysis of the predictor-corrector scheme as a theoretical support is provided in Appendix Section C.3.

**T1 Processor:** In this part, we regard the outputs of each S-block $\phi_s^l$ as estimated solutions at certain intermediate time points. Specifically, given an input $\mathbf{h}^l$, the predictor produces $\mathbf{h}^{l+1,*}$ as defined in Eq. 8. To refine this prediction, we introduce a learnable block $\phi_t^l$ that corrects $\mathbf{h}^{l+1,*}$ to obtain the final solution $\mathbf{h}^{l+1}$, denoted as $\phi_t^l : (\mathbf{h}^{l+1,*}, \mathbf{h}^l) \mapsto \mathbf{h}^{l+1}$. Concretely, we (1) construct a temporal affine mapping $\phi_a^l$ between $\mathbf{h}^{l+1,*}$ and $\mathbf{h}^l$, i.e., $\phi_a^l : (\mathbf{h}^{l+1,*}, \mathbf{h}^l) \mapsto \mathbf{A}^l$, serving as a lightweight surrogate for $\phi_s^l$ in Eq. 8; and (2) apply this mapping on $\mathbf{h}^{l+1,*}$ to obtain the corrected features $\mathbf{h}^{l+1}$, described as follows:

$$\mathbf{h}^{l+1} = \phi_m^l\big(\mathbf{h}^l + \mathbf{A}^l\big(\mathbf{h}^{l+1,*}\mathbf{W}_3^l\big)\big) \quad \text{with} \quad \mathbf{A}^l = (\mathbf{h}^{l+1,*}\mathbf{W}_1^l)(\mathbf{h}^l\mathbf{W}_2^l)^T \tag{9}$$

where $\mathbf{A}^l \in \mathbb{R}^{N \times N}$ is the learned transformation matrix, $\mathbf{W}_1^l \in \mathbb{R}^{d_l \times d_l}$, $\mathbf{W}_2^l \in \mathbb{R}^{d_l \times d_l}$, and $\mathbf{W}_3^l \in \mathbb{R}^{d_l \times d_l}$ are the learnable weights, $\phi_m^l$ is a differentiable function (e.g., MLPs). After several rounds of iteration within the processor, the output of last T-block is then fed into the Decoder for final dynamics prediction.

*Remark* 3.3 (**The correct direction of state transmission in temporal strategy**). In practice, our method is essentially designed to conform a physically consistent direction of temporal state transmission. Due to the error accumulation over time, the state from the previous timestep is generally more reliable than the current predicted one (Bathe, 2007), which is commonly observed in autoregressive modeling. This rationale also explains why the causal attention mechanism with Markov assumptions is more suitable for the multi-input temporal learning (Yang et al., 2021). Therefore, our strategy consistently uses the previous state to guide the evolution of the current state, avoiding the use of potentially noisy current states to influence past states. Detailed results supporting this analysis are provided in Table 4.

## 4 EXPERIMENTS

**Datasets:** To evaluate whether DSTGN provide a greater generalization capability on complex spatiotemporal dynamics through decoupled learning, we consider two test conditions: different spatial dimensions and varying temporal scales. To this end, we selected various systems from publicly available datasets: (1) Basic PDE systems, which includes the 2D Burgers equation and the 2D and 3D Gray-Scott (GS) systems, (2) 2D Black Sea (BS) dataset for oceanographic forecasting, and (3) 2D Unmanned Aerial Vehicle Maneuvering (UAVM) dataset. PDE systems from (Rao et al., 2023; Kochkov et al., 2021) are spatially and temporally down-sampled (e.g., 2-fold in space and 5-fold in time) from high-resolution data generated using numerical solvers under different initial conditions (ICs) and periodic boundary conditions (BCs), while the BS dataset was collected from years of field measurements, which may contain substantial noise and systematic errors. Moreover, the UAVM dataset simulates the complex airflow patterns of the unmanned aerial vehicle maneuvering in the varying environments. More details are shown in Appendix Section E.

**Scenarios:** In summary, the spatial scale includes 2D and 3D space, while the temporal interval ranges from 0.001 seconds to one day. Importantly, unlike the static mesh in other systems, the UAVM dataset utilizes the dynamic mesh (the number and location of nodes are different at varying times in each evolution), which poses a big challenge to the generalization ability of surrogate models.

**Baselines:** To quantitatively evaluate the performance improvement of DSTGN, we conducted comparisons with various baselines (for further details, see Appendix Section F), including GAT (Velikovi et al., 2018), GATv2 (Brody et al., 2022), MeshGraphNet (MGN) (Pfaff et al., 2021), Message-Passing PDE Solver (MP-PDE) (Brandstetter et al., 2022), FNO (Li et al., 2021), FFNO (Tran et al., 2021), GeoFNO (Li et al., 2024a), OFormer (Li et al., 2023), and Transolver (Wu et al., 2024). Here, OFormer model include two patterns: OFormer-F (OFF) with the "fourier" attention and OFormer-G (OFG) with the "galerkin" attention. The hyperparameters are listed in Appendix Section G.

**Loss function:** To minimize the discrepancy between the labeled and predicted data, we adopt the following loss function in the training procedure, defined as $\mathcal{L}(\boldsymbol{\theta}) = \frac{1}{N} \sum_{\alpha=1}^{N} \|\mathbf{u} - \hat{\mathbf{u}}\|_2^2$, where $N$ is the number of spatial sample points, $\hat{\mathbf{u}} \in \mathbb{R}^{N \times d_v}$ the predicted data and $\mathbf{u} \in \mathbb{R}^{N \times d_v}$ the truth labels. The value of rooted mean square errors (RMSE) acts as the evaluation metric.

**Training settings:** In our work, we train each model three times and primarily adopt the one-step training strategy. In addition, we train all models using the Adaptive Moment Estimation (Adam) optimizer (Kingma & Ba, 2015) and the ReduceLROnPlateau learning scheduler (Paszke et al., 2019) with a decay of $0.8$. For fairness, the latent feature dimension of all model is same. Additionally, we utilize layer normalization to improve the training convergence, except for the final layer in the decoder block. The GELU activation function is applied in all MLPs of our model. More details about the hyperparameters are given in Appendix Section D. In addition, all experiments are run on NVIDIA A100 GPU. The input of all models have been injected random Gaussian noise of varying standard deviation to improve stability and accuracy. More details refer to Appendix Section G.

### 4.1 STATIC MESH SYSTEM

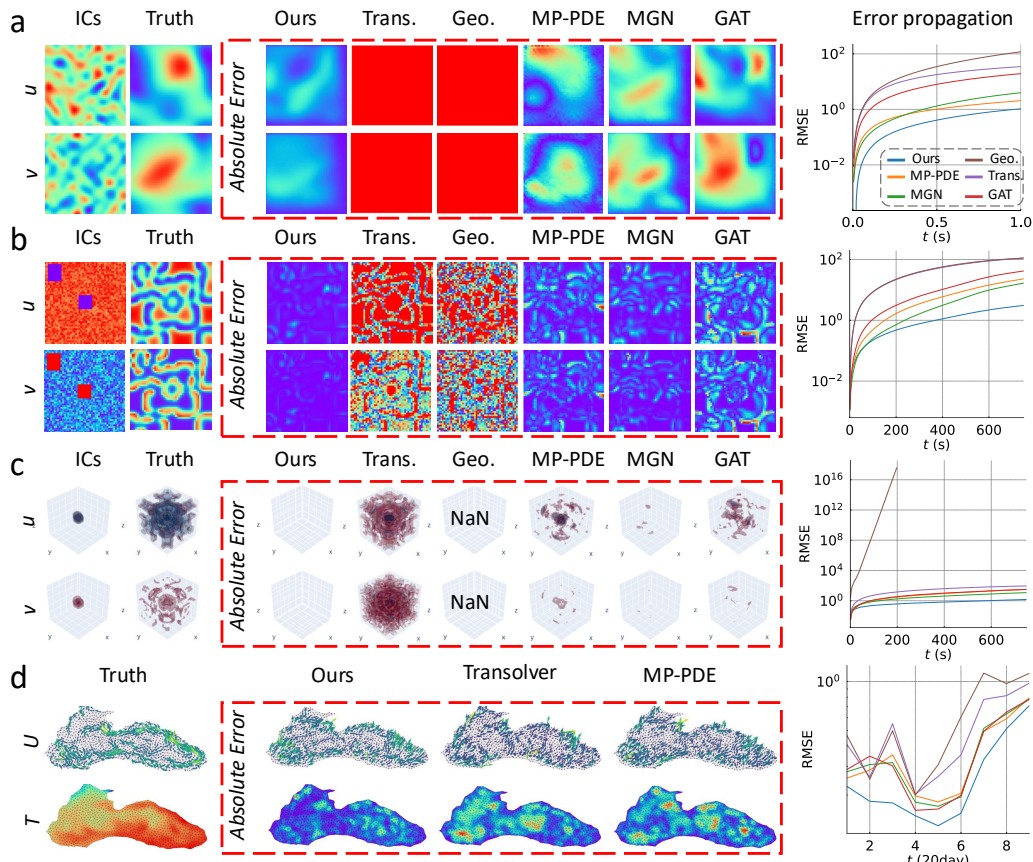

Figure 4: Visualization snapshots and error propagation on the static mesh datasets. Here, the area enclosed by the red dotted line represents the snapshots of absolute error between ground-truth data and the prediction values. The abbreviation "Geo." and "Trans." represent the GeoFNO model and Transolver model. Full evolutions refer to Appendix Figures S2 to S8. The time range of error propagation of several systems is varying. For example, the case of Burgers equation is $t \in [0, 1]s$, 2D and 3D GS equations $t \in [0, 750]s$, and 2D BS dataset every 20-day period (total 9 periods).

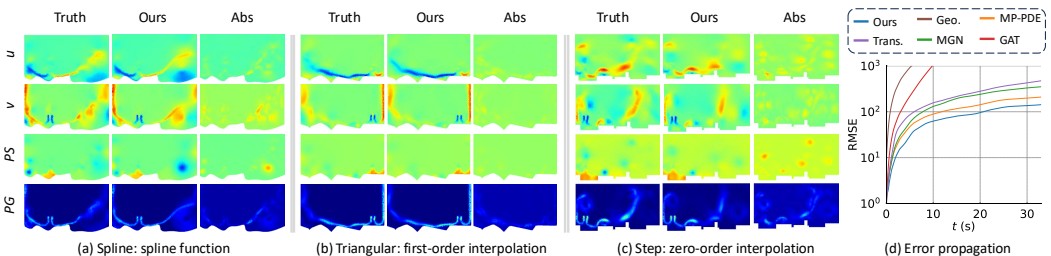

Figure 5: Visualization snapshots of DSTGN on varying floors. **a**, the case of the spline function. **b**, the case of the first-order interpolation function. **c**, the case of the zero-order interpolation function. **d**, the error propagation of UAVM dataset at $t \in [0, 33]s$.

**Basic PDE systems:** Table 1 presents the comparative performance of all models on PDE systems. The results show that DSTGN outperforms other baselines under the RMSE metric. A reduction of one order of magnitude in RMSE indicates that our model can learn the dynamics well with a great generalization capability. As shown in Figure 4, we provide the snapshots of generalization tests for all baselines at the final time step (e.g., 1.0 s of Burgers equation, 750.0 s of

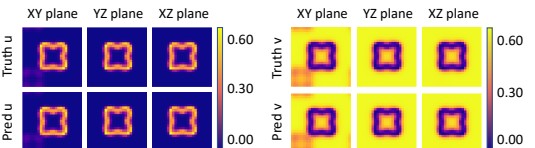

Figure 6: Plane slices of the prediction results of DSTGN on one 3D Gray-Scott dataset at 300 s.

GS RD equations). For clarity, we provide the prediction error propagation curves of all models are displayed in Figures 4 and 5. Our analysis indicates that all graph-based methods could learn a basic pattern while some models (e.g., GAT) do not perform well. Notably, the Transformer model and its variants also do not exhibit considerable performance in all benchmarks, even the simple Burgers equation. This phenomenon indicates that the attention mechanism is not well developed to learn complex dynamics without the help of the underlying ingenious temporal strategy. Unexpectedly, GeoFNO shows the NaN phenomenon in the prediction process on some 3D test cases, whose RMSE value reach $10^{15}$ at 200.0 s. This reflects its instability in challenging data-scarce or long-term prediction settings. Transolver also lacks the capability to learn the intricate evolution of the 3D system states in the regime of small training data, resulting in substantial errors. Visualizations of the flow evolution under a randomized IC, as predicted by our model, are provided in Appendix Figures S4, S5, and S6. We also provide the plane slices of our model's prediction on 3D GS system at 300.0 s in Figure 6. More results and analyses are available in Appendix Section H.

**BS dataset:** As shown in Figure 4**d**, we exhibit nine 20-day-long predictions of water flow velocities and sea surface temperature over a 180-day period on the Black Sea. It is evident that our model generalizes well with lower error values on all 9 periods. A notable observation from Figure 4**d** is that temperature predictions are more accurate than those for flow velocity. This discrepancy may be attributed to the relative stability of water temperature at a depth of 12 meters, where the influence of solar radiation is diminished, resulting in relatively stable water temperatures and less prone to dramatic fluctuations. Consequently, predicting flow velocity likely requires a broader set of reference variables. However, the available dataset comprises only the flow velocity and water temperature, which may explain this performance gap. Despite the clear superiority over other baselines (see Figure 4**d**), we would like to believe that incorporating more types of related variables into our model could further enhance the model's capability. A visualization of flow evolution under

Table 1: Results of different methods on various benchmarks. Here, "–" represents that the model is unable or unsuitable to learn the dynamics directly. "↓" represents that the smaller the value of the quantitative metric, the better the model performance. The **bold** values and underlined values represent the optimal and sub-optimal results on various datasets. Abbreviation "Pro." is the promotion, calculated from the above two.

| Model | Regular ↓ | | | Irregular ↓ | |
|---|---|---|---|---|---|
| | Burgers | 2D GS | 3D GS | BS | UAVM |
| MLP | 0.1798 | 0.2873 | 0.2907 | 0.6812 | 0.6726 |
| GAT | 0.1175 | 0.0722 | 0.0639 | 0.6295 | 1.3063 |
| GATv2 | 0.1194 | 0.0730 | 0.0451 | 0.6479 | 1.4251 |
| MGN | 0.0117 | 0.0291 | 0.0192 | 0.6147 | 0.4420 |
| MP-PDE | 0.0178 | 0.0386 | 0.0652 | 0.6076 | 0.3651 |
| FNO | 0.0575 | 0.1133 | 0.1716 | – | – |
| FFNO | 0.0334 | 0.0362 | 0.0359 | – | – |
| GeoFNO | 0.5936 | 0.1866 | NaN | 1.2893 | NaN |
| Transformer | 0.0374 | 0.1414 | 0.1178 | 0.5583 | 0.6856 |
| OFormer-F | 0.1080 | 0.1903 | 0.1531 | 0.7616 | 0.8544 |
| OFormer-G | 0.1505 | 0.1894 | 0.1518 | 0.7682 | 0.8563 |
| Transolver | 0.1742 | 0.1859 | 0.1520 | 0.8199 | 0.5957 |
| Ours | **0.0065** | **0.0037** | **0.0089** | **0.5059** | **0.2784** |
| Pro. (%) ↑ | 44.4 | 87.2 | 53.6 | 9.38 | 23.74 |

a randomized IC, as predicted by our model, is provided in Appendix Figure S7. Further details are available in Appendix Section H.

## 4.2 DYNAMIC MESH SYSTEM

**UAVM dataset:** As shown in Figure 5, we provide the snapshots of our model's predictions on three distinct types of floor profiles, reflecting a great generalization capability. Given the predicted results and corresponding error distributions, it is evident that our model effectively captures the dynamic evolution induced by UAV flight. However, while the low-frequency modes are well represented, high-frequency errors remain. This issue will be the focus of our future research and we would continue to optimize our model. A quantitative summary of our model and all baselines on 2D UAVM dataset is provided in Table 1. As expected, Transolver achieves competitive performance when trained on large datasets. Moreover, both MGN and MP-PDE maintain robust performance, confirming their representativeness as strong graph-based baselines. With the help of local spatial modeling, graph-based models perform well, except for the family of GAT, which relies solely on the attention mechanism and abandons edge features. Additional prediction snapshots of our model are provided in Appendix Figure S8, with further details in Appendix Section H.

## 4.3 ABLATION STUDY

To evaluate the contributions of the S-block and T-block, we perform an ablation study on all benchmarks. As summarized in Table 2, our model with decoupled learning shows better generalization capability. While the degree of improvement varies, both blocks contribute significantly across differ-

ent tasks. Our proposed temporal strategy also enhances the stability and accuracy of DSTGN, even in the case of limited training data. A substantial performance gap is observed between our model and all baselines across benchmarks, including scenarios with small time intervals (e.g., 0.001 s for the 2D Burgers equation). Extended ablation studies refer to Appendix Section H.

As shown in Table 3, the results validate the rationality of our architectural design. While varying learning patterns exhibit different performances, our architecture consistently outperforms the alternative configurations. We further verify that our proposed temporal strategy is not merely a simple modification of the temporal transmission sequence. As detailed in Table 4, which encompasses tests across all datasets, the results corroborate our temporal hypothesis and confirm that the learnable strategy effectively mitigates error accumulation.

**Effect of injecting training noise:** In our work, we consider a Gaussian noise injection strategy (Sanchez-Gonzalez et al., 2020; Pfaff et al., 2021) in training stage for stabilizing the model and enhancing the generalization ability. An ablation study on this strategy (see Appendix Table S5) demonstrates that injecting training noise leads to consistent performance improvements.

**Comparison under controlled conditions:** To further ensure a fair comparison, we conduct additional experiments under controlled conditions, matching either the parameter volume (approximately 2 million parameters) or the inference time (0.008 s per step) of competing methods. The results, provided in Appendix Table S6, consistently support the effectiveness and superiority of our method over other baselines.

Table 2: Ablation study of DSTGN on all datasets. Abbreviations "S1","S2", and "T1" represent the S1, S2 and T1 processors.

| Model | Burgers | 2D GS | 3D GS | BS | UAVM |
|---|---|---|---|---|---|
| S1 | 0.0117 | 0.0291 | 0.0192 | 0.6147 | 0.4420 |
| S1 + S2 | 0.0077 | 0.0079 | 0.0097 | 0.5287 | 0.3653 |
| S1 + T1 | 0.0085 | 0.0071 | 0.0101 | 0.5170 | 0.3279 |
| S2 + T1 | 0.0207 | 0.0748 | 0.0612 | 0.5359 | 0.6127 |
| Ours | **0.0065** | **0.0037** | **0.0089** | **0.5059** | **0.2784** |

Table 3: Quantitative results of various architecture designs. Abbreviations "S", "T", and "T*" represent the S-block, T-block, and T-block only handling current temporal states.

| Case | Burgers | 2D GS | 3D GS | BS | UAVM |
|---|---|---|---|---|---|
| $S \rightarrow T^*$ | 0.0086 | 0.0074 | 0.0109 | 0.5247 | 0.3450 |
| $T^* \rightarrow S$ | 0.0088 | 0.0091 | 0.0112 | 0.5259 | 0.3707 |
| $S \parallel T^*$ | 0.0083 | 0.0053 | 0.0125 | 0.5152 | 0.3354 |
| Ours ($S \rightarrow T$) | **0.0065** | **0.0037** | **0.0089** | **0.5059** | **0.2784** |

Table 4: Quantitative results of varying flow directions of T-block within DSTGN on all datasets. Abbreviations "I" and "O" represent the input and output of the former S-block. Function $\phi_a$ belongs to T-block in Eq. 9.

| Flow direction | Burgers | 2D GS | 3D GS | BS | UAVM |
|---|---|---|---|---|---|
| $(I, I) \rightarrow \phi_a \rightarrow I$ | 0.0083 | 0.0053 | 0.0125 | 0.5152 | 0.3354 |
| $(O, O) \rightarrow \phi_a \rightarrow O$ | 0.0086 | 0.0074 | 0.0109 | 0.5247 | 0.3450 |
| $(I, O) \rightarrow \phi_a \rightarrow O$ | 0.0085 | 0.0070 | 0.0117 | 0.5213 | 0.3491 |
| $(O, I) \rightarrow \phi_a \rightarrow I$ | 0.0088 | 0.0079 | 0.0116 | 0.5358 | 0.3701 |
| $(I, O) \rightarrow \phi_a \rightarrow I$ | 0.0089 | 0.0082 | 0.0128 | 0.5419 | 0.3720 |
| $(O, I) \rightarrow \phi_a \rightarrow O$ | **0.0065** | **0.0037** | **0.0089** | **0.5059** | **0.2784** |

**Running time and memory information:** Given above setups, we provide the training time, the inference time and the memory overhead in Appendix Table S7. Also, we provide an ablation study to show our model's efficiency when the temporal module is removed. Notably, without the temporal module, our approach becomes more efficient while still achieving competitive results.

**Hyperparameter test:** We provide the hyperparameter (e.g., hidden layer, Std. of noise, and $k$ value in S2 Processor) sensitivity study in Appendix Tables S8-S10. The results for node scaling test are also listed in Appendix Table S11.

## 5 CONCLUSION

In this paper, we propose a provably powerful spatiotemporal framework (i.e., DSTGN) to model the complex dynamics on both regular and irregular domains. DSTGN enhances spatial representation learning by explicitly capturing channel-wise interactions in latent feature spaces, while its novel learnable temporal integration mechanism guides layer-wise feature evolution and mitigates error accumulation in autoregressive inference. The effectiveness of DSTGN has been demonstrated through extensive experimental results on various benchmarks, including three synthetic PDE systems, a real-world dataset, and a complex unmanned aerial vehicle maneuvering simulation. Currently, DSTGN achieves a learnable generalization of the predictor–corrector scheme in the context of a neural network framework. A promising direction for future work is to extend this approach to higher-order schemes, which are theoretically capable of achieving greater accuracy, albeit at the cost of increased computational complexity. We plan to systematically explore this trade-off and address the associated challenges in subsequent research.

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

# APPENDIX

## LLM USAGE STATEMENT

In accordance with the conference policy on the use of Large Language Models (LLMs), we declare that LLMs were used solely as an auxiliary tool for checking grammar, improving language clarity, and polishing the readability of the manuscript. No part of the research ideation, methodological design, experimental execution, or substantive writing relied on LLM assistance. The authors take full responsibility for the content of this paper.

## IMPACT STATEMENT

This paper presents an end-to-end graph model for learning complex dynamical systems, e.g., climate systems, chemical reaction-diffusion systems, turbulent flows, etc., across various disciplines. This work focuses on developing new learning approaches and possesses no ethical risks.

## A   NOTATION

In this part, we present a summary of standardized notations, as detailed in Table S1.

## B   SPACE CONCEPT

In this part, we briefly introduce the concept of topological simplex on graphs.

**Simplex and Complex:** The simplex and complex generalize the notion of triangle or tetrahedron in geometry to arbitrary dimensions, which is defined as follows.

**Definition B.1** (Simplex). A $k$-simplex $\sigma_k$ is a $k$-dimensional polytope, which is the convex hull of its $k + 1$ affinely independent vertices.

**Definition B.2** (Complex). Geometrically, a simplicial complex $\mathcal{K}$ is a finite set of simplices that satisfies the face and intersection conditions: (1) Every face of a simplex in $\mathcal{K}$ is also in $\mathcal{K}$ and (2) The intersection of any two simplices $\sigma_1, \sigma_2 \in \mathcal{K}$ is either empty or a shared face.

Based on the above theories, we extend the topological knowledge to graph space. Generally, a 0-dimensional simplex is a *vertice*, a 1-dimensional simplex is an *edge*, a 2-dimensional simplex is a *triangle*, and so on. We provide the corresponding definitions on graphs.

**Definition B.3.** Consider a 0-simplex (vertice) $\sigma \in \mathcal{K}$, there are four types of simplices within every 1-hop adjacent neighborhood: (1) 0-simplex $\sigma_0$, which is a neighbor of 0-simplex $\sigma$ (namely, vertices), (2) 1-simplex $\sigma_1$, which contains $\sigma$ and $\sigma_0$ (namely, edge), (3) $k$-simplex $\sigma_{k,\in}(k \geq 2)$, which contains $\sigma$, and (4) $k$-simplex $\sigma_{k,\notin}(k \geq 2)$, which does not contain $\sigma$, but is spatially adjacent to $\sigma$.

## C   TIME INTEGRATION

Autoregressive prediction is prone to error accumulation, where even small prediction errors gradually amplify over time as the model relies on its own predictions for future steps within a discrete time span. The time integration methods are proposed to address this fundamental limitation through the multi-stage temporal continuity approximation strategy. Generally, the time integration includes the explicit and implicit forms, defined as follows.

**Definition C.1** (Explicit time integration). Consider a differential equation $du/dt = f(u,t)$, the simplest explicit scheme updates the solutions via the current states at time $t_l$, described as:

$$u^{l+1} = u^l + \Delta t \cdot f(u^l, t_l), \tag{S1}$$

where $u^l$ is the solution at time $t_l$, $\Delta t$ is the time step size, $f(u^l, t_l)$ defines the rate of change.

Table S1: Summary of Notations.

| Calculus | Short Name |
|---|---|
| Derivative of $y$ with respect to $x$ | $dy/dx$ |
| Partial derivative of $y$ with respect to $x$ | $\partial y/\partial x$ |
| Gradient of $y$ with respect to $x$ | $\nabla_x y$ |
| Tensor containing derivatives of $y$ with respect to $\mathbf{X}$ | $\nabla_{\mathbf{X}} y$ |
| Jacobian matrix $\mathbf{J} \in \mathbb{R}^{m \times n}$ of $f : \mathbb{R}^n \to \mathbb{R}^m$ | $\partial f/\partial x$ |
| The Hessian matrix of $f$ at input point $x$ | $\nabla_x^2 f(x)$ or $\mathbf{H}(f)(x)$ |
| Definite integral over the entire domain of $x$ | $\int f(x)dx$ |
| Definite integral with respect to $x$ over the set $S$ | $\int_S f(x)dx$ |

| Sets and Graphs | Short Name |
|---|---|
| The set containing 0 and 1 | $\{0, 1\}$ |
| The set of all integers between 0 and $n$ | $\{0, 1, \ldots, n\}$ |
| The real interval including $a$ and $b$ | $[a, b]$ |
| The real interval excluding $a$ but including $b$ | $(a, b]$ |
| A graph with nodes and edges | $G = (E, V)$ |

| Latent Variables | Short Name |
|---|---|
| The node index | $i, j, k, r$ |
| The latent node features | $\mathbf{h}_i$ |
| The latent edge features | $\mathbf{e}_{ij}$ |
| The latent cell features | $\mathbf{c}_{ijk}$ |
| The node functions | $\phi_v$ |
| The edge functions | $\phi_e$ |
| The cell functions | $\phi_c$ |
| The graph functions | $\phi_g$ |
| The transformer functions | $\phi_t$ |

| Variables | Short Name |
|---|---|
| x-component of velocity $\mathbf{u}(\mathbf{x}, t)$ | $u$ |
| y-component of velocity $\mathbf{u}(\mathbf{x}, t)$ | $v$ |
| vorticity | $w(\mathbf{x}, t)$ |
| pressure | $p(\mathbf{x}, t)$ |
| temperature under water | $T(\mathbf{x}, t)$ |

| Space and Time | Short Name |
|---|---|
| x-direction of space coordinate | $x$ |
| y-direction of space coordinate | $y$ |
| z-direction of space coordinate | $z$ |
| time coordinate | $t$ |
| time increment | $\Delta t$ |
| space increment | $\Delta x$ |
| discrete timestamp at $k$th step | $t_k$ |

Table S2: Explicit time integration and implicit time integration.

| Time integration | Explicit | Implicit |
|---|---|---|
| Time step size | small | large |
| Computational cost | low | high |
| Stability | unstable | stable |
| Applications | non-stiff systems | stiff systems |

**Definition C.2** (Implicit time integration). Consider a differential equation $du/dt = f(u, t)$, the simplest implicit scheme updates the solutions via the current states at time $t_l$ and the next states at time $t_{l+1}$, described as:

$$u^{l+1} = u^l + \Delta t \cdot f(u^{l+1}, t_{l+1}), \tag{S2}$$

where $u^{l+1}$ appears on both sides of the equation.

## C.1 Common methods

We briefly introduce several time integration methods: Finite Difference scheme and Runge-Kutta method. Generally, temporal approaches directly predict the next states based on previous states with the autoregressive form:

$$\mathbf{u}(t + \Delta t) = \phi_t \left( \mathbf{u}(t) \right) \tag{S3}$$

where $\mathbf{u}(t)$ represents the state at time $t$, $\Delta t$ is the time interval, $\phi_t(\cdot)$ is the function for prediction. This basic approach constructs the basic temporal transmission in the training and inference stages.

In the following, we firstly focus on first-order finite difference methods, including (1) Forward Euler method and (2) Central Difference method. For more advanced methods, refer to the book (Stoer et al., 1980).

**Forward Euler method:** Unlike the above method learning the next states directly, Forward Euler method predicts the next states with an increment learning:

$$\mathbf{u}(t + \Delta t) = \mathbf{u}(t) + \Delta t \cdot \phi_t(\mathbf{u}(t)) \tag{S4}$$

where $\phi_t(\cdot)$ is utilized to learn the increment based on the previous states. This approach increases the efficiency of training and inference with first-order error precision guarantee.

**Central Difference method:** Central Difference method predicts the next states with the multiple previous states. For clarity, first-order central difference method learns from the previous two states, defined in the following form:

$$\mathbf{u}(t + \Delta t) = \mathbf{u}(t - \Delta t) + 2\Delta t \cdot \phi_t(\mathbf{u}(t)) \tag{S5}$$

where the method considers both forward and backward time steps at time $t$, improving accuracy when the evolution is relatively smooth over time.

**Runge-Kutta method:** The Runge-Kutta method is the most popular time integration schemes in physics. For a differentiable variable $\mathbf{u}$, there is a function $d\mathbf{u}/dt = \phi_t(t, \mathbf{u})$. We can solve above equation with the classic fourth-order Runge-Kutta (RK4) method based on four intermediate states (i.e., $k_1, k_2, k_3$ and $k_4$):

$$k_1 = \phi_t(t, \mathbf{u}(t)) \tag{S6a}$$
$$k_2 = \phi_t(t + \Delta t/2, \mathbf{u}(t) + (\Delta t/2)k_1) \tag{S6b}$$
$$k_3 = \phi_t(t + \Delta t/2, \mathbf{u}(t) + (\Delta t/2)k_2) \tag{S6c}$$
$$k_4 = \phi_t(t + \Delta t, \mathbf{u}(t) + \Delta t k_3) \tag{S6d}$$
$$\mathbf{u}(t + \Delta t) = \mathbf{u}(t) + \Delta t/6(k_1 + 2k_2 + 2k_3 + k_4) \tag{S6e}$$

where the method offers superior accuracy along with a significantly increased computational cost. Multiple intermediate states make the method unstable in long-term prediction, significantly magnifying the error accumulation issue. We do not adopt this type of method in our work due to their instability and 4-fold cost of GPU usage and computation time.

## C.2 Our temporal strategy

**Overview:** The above limitations motivate us to propose the learnable time integration scheme, which provides a more practical and robust approach for long-term prediction. Inspired by the Predictor-Corrector strategy (Wanner & Hairer, 1996), a numerical method for solving differential equations (see below for detailed procedures), we combine explicit (predictor) and implicit (corrector) approaches to construct a temporal strategy within the network transmission process. The predictor step provides an initial estimate of the solution at the next time step using an explicit method, while the corrector step refines this estimate by incorporating implicit computations to improve accuracy and stability.

**Predictor Step**: Use an explicit method to predict the solution at the next time step.

$$\bar{\mathbf{u}}(t + \Delta t) = \mathbf{u}(t) + \Delta t \cdot \phi_t(\mathbf{u}(t)) \tag{S7}$$

**Corrector Step**: Use an implicit method to correct the predicted value.

$$\hat{\mathbf{u}}(t + \Delta t) = \frac{\Delta t}{2}(\phi_t(\mathbf{u}(t)) + \phi_t(\bar{\mathbf{u}}(t + \Delta t))) \tag{S8a}$$

$$\mathbf{u}(t + \Delta t) = \mathbf{u}(t) + \hat{\mathbf{u}}(t + \Delta t) \tag{S8b}$$

This method combines the computational efficiency of explicit methods and the robustness of implicit methods, achieving second-order accuracy or higher, albeit at the cost of increased computational requirements due to the need to handle multiple intermediate variables

## C.3 THEORETICAL DERIVATION OF THE LOCAL TRUNCATION ERROR (LTE)

Consider the initial value problem $dy/dt = f(t, y(t)), y(t_0) = y_0$ where $y_n \approx y(t_n)$ is known, we aim to compute the predicted value $y_{n+1}$ at time $t_{n+1}$ (i.e., $t_n + \Delta t$), where $\Delta t$ is the time step size.

*Proof.* Using Taylor expansion, we express the exact solution at time $t_{n+1}$ as:

$$y(t_{n+1}) = y(t_n) + (\Delta t)y'(t_n) + \frac{(\Delta t)^2}{2}y''(t_n) + \frac{(\Delta t)^3}{6}y'''(t_n) + \mathcal{O}((\Delta t)^4), \tag{S9}$$

$$y'(t_n) = f(t_n, y_n), \quad y''(t_n) = f_t + f_y y' = f_t(t_n, y_n) + f_y(t_n, y_n)f(t_n, y_n), \tag{S10}$$

where higher-order terms $y'(t_n), y''(t_n)$, and $y'''(t_n)$ represent successive derivatives of $f$. To simplify the analysis, we retain only the first four terms and omit the detailed expansion of $y'''(t_n)$.

Following the Predictor-Corrector scheme, we have:

$$\begin{aligned} y_{n+1} &= y_n + \frac{(\Delta t)}{2}\left[f(t_n, y_n) + f(t_{n+1}, \tilde{y}_{n+1})\right] \\ &= y_n + \frac{(\Delta t)}{2}\left[2f(t_n, y_n) + (\Delta t)f_t + (\Delta t)f_y f + \mathcal{O}((\Delta t)^2)\right] \\ &= y_n + (\Delta t)f(t_n, y_n) + \frac{(\Delta t)^2}{2}(f_t + f_y f) + \mathcal{O}((\Delta t)^3) \end{aligned} \tag{S11}$$

Subtracting the numerical approximation from the exact solution yields:

$$y(t_{n+1}) - y_{n+1} = \frac{(\Delta t)^3}{6}y'''(t_n) + \mathcal{O}((\Delta t)^4) \tag{S12}$$

Thus, its local truncation error is $\mathcal{O}((\Delta t)^3)$, confirming the second-order accuracy. Similarly, following the explicit time integration scheme (e.g., forward Euler method), we have the local truncation error of $\mathcal{O}((\Delta t)^2)$. Our scheme is more stable than standalone explicit methods. It avoids the high computational cost of fully implicit methods due to the only requirement of the real variables at the current moment. □

In summary, we have explained how our method impact the accuracy and stability of the temporal evolution. The approach can be extended naturally to higher-order learnable schemes (e.g., replacing the explicit scheme with Runge-Kutta method) to further optimize precision and robustness. As more intermediate nodes are inserted, our method enforces derivative consistency, ensuring smoother temporal transmission and the improved solution quality.

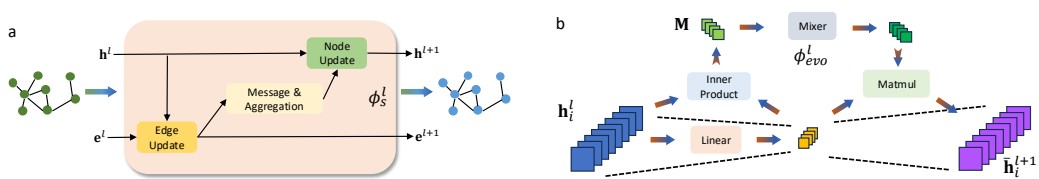

Figure S1: Supplementary flow diagram of our architecture in Figure 1. **a**, the diagram of S1 Processor. **b**, the diagram of S2 Processor.

---

**Algorithm 1 D**ecoupled **S**patio**T**emporal **G**raph **N**etwork (DSTGN)

---

**Input:** Node features $\mathbf{h}_i^l$; Edge features $\mathbf{e}_{ij}^l$;
**Function:** Node function $\phi_v^l$; Edge function $\phi_e^l$; Temporal block $\phi_t^l$; Evolutional block $\phi_{evo}^l$;
**Parameter:** Weights $\mathbf{W}_0^l$, $\mathbf{W}_1^l$, $\mathbf{W}_2^l$, and $\mathbf{W}_3^l$;
**Output:** Node features $\mathbf{h}_i^{l+1}$; Edge features $\mathbf{e}_{ij}^{l+1}$.

---

1: **while** stop condition is not reached **do**
2:    **for** node $\mathbf{h}_i^l$ and edge $\mathbf{e}_{ij}^l$ within S1 Processor **do**
3:       Compute the new edge features: $\mathbf{e}_{ij}^{l+1} \leftarrow \phi_e^l \left( \mathbf{h}_i^l \parallel \mathbf{h}_j^l \parallel \mathbf{e}_{ij}^l \right)$;
4:       Aggregate edges and Update node features: $\check{\mathbf{h}}_i^{l+1} \leftarrow \phi_v^l \left( \mathbf{h}_i^l \parallel \sum_{j \in \mathcal{N}_i} \mathbf{e}_{ij}^{l+1} \right)$;
5:    **end for**
6:    **for** all node $\mathbf{h}^l$ within S2 Processor **do**
7:       Compute the first mapping: $\mathbf{M} \leftarrow (\mathbf{h}^1 \mathbf{W}_0^1) \cdot \mathbf{h}^1$;
8:       Compute node features with evolutional mapping: $\bar{\mathbf{h}}^{l+1} \leftarrow (\mathbf{h}^l \mathbf{W}_0^l) \left( \phi_{evo}^l (\mathbf{M}) \right)$;
9:    **end for**
10:   Obtain the output of S-block: $\mathbf{h}^{l+1,*} \leftarrow \check{\mathbf{h}}^{l+1} + \bar{\mathbf{h}}^{l+1}$;
11:   **for** all node $\mathbf{h}^l$ within T1 Processor **do**
12:      Learn the next intermediate states: $\mathbf{h}^{l+1} \leftarrow \mathbf{h}^l + \phi_t^l(\mathbf{h}^{l+1,*}, \mathbf{h}^l))$;
13:   **end for**
14: **end while**
15: return the next states $\mathbf{h}_i^{l+1}$ and $\mathbf{e}_{ij}^{l+1}$.

---

## D  IMPLEMENTATION DETAILS

**Pseudo Code:** As shown in Algorithm 1, we provide the detailed procedure of DSTGN. If we adopt the graph network in S1 Processor, the overall complexity is $O((|V| + |E|)d^2 + |V|^2 d)$, where $|V|$ and $|E|$ are the number of nodes and edges.

**Graph Network in S1 Processor:** Graph Networks (Gilmer et al., 2017a) model the interactions between nodes (vertices) in graphs by iteratively passing "message" along edges and updating node representations based on these messages. Consider the graph $G = (V, E)$, where $V$ is the set of nodes (vertices), $E$ is the set of edges, $\mathbf{h}_i$ represents the feature of node $i$, $\mathbf{e}_{ij}$ represents the edge feature between nodes $i$ and $j$. As shown in Figure S1**a**, the key components include that: (1) **Passing Message**: Nodes exchange the message $\mathbf{e}_{ij}$ from node $i$ to node $j$ computed by a function: $\mathbf{e}_{ij}^{l+1} = \phi_e^l(\mathbf{h}_i^l, \mathbf{h}_j^l, \mathbf{e}_{ij}^l)$; (2) **Aggregation**: Nodes aggregate upstream messages using a permutation-invariant function (e.g., summation, mean, max): $\mathbf{m}^{l+1} = \sum_{j \in \mathcal{N}_i} \mathbf{e}_{ij}^{l+1}$, where $\mathcal{N}_i$ denotes the neighbor set of node $i$; (3) **Updating Node**: Nodes update the states using the function $\phi_v$ (e.g., a multi-layer perceptron): $\mathbf{h}_i^{l+1} = \phi_v^l(\mathbf{h}_i^l, \mathbf{m}^{l+1})$; (4) **Readout**: Applying a readout function to obtain the node representations or the graph structure, depending on the task (e.g., node classification, graph classification): $\hat{\mathbf{u}}_i = \phi^{de}(\mathbf{h}_i^L)$, where $\phi^{de}$ is a function to decode the latent features to physical variables and $L$ is the number of the iterations.

**Mixer block in S2 Processor:** For the input $\mathbf{h} \in \mathbb{R}^{n \times d}$, where $n$ is the number of tokens (or patches), and $d$ is the embedding dimension of each token, the computation in a Mixer model in Figure S1**b** typically involves the following steps: (1) **Token Mixing**: The first block mixes information

Table S3: Basic information of datasets. The trajectory number for training, validation, and testing is described by the form like (5/2/3).

| Dataset | Domain | Physical parameters | No. of nodes | Trajectory length | Train/Validation/Test trajectories | Boundary condition | Force term |
|---|---|---|---|---|---|---|---|
| 2D Burgers | Square $[0,1]^2$ | $(u,v)$ | 2,500 ($50^2$) | 1,000 | 40 (30/5/5) | Periodic | No |
| 2D GS | Square $[0,96]^2$ | $(u,v)$ | 2304 ($48^2$) | 3,000 | 10 (5/2/3) | Periodic | No |
| 3D GS | Cubic $[0,96]^3$ | $(u,v)$ | 13,824 ($24^3$) | 3,000 | 6 (2/2/2) | Periodic | No |
| 2D BS | Irregular | $(u,v,T)$ | 1,000∼40,000 | 365 | 24 (20/2/2) | Unknown | Unknown |
| 2D UAVM | Irregular | $(u,v,ps,pg)$ | 3,000∼5000 | 990 | 1184(948/118/118) | Unknown | Unknown |

across tokens (spatial dimension) while keeping the channel dimension constant, expressed as: $\mathbf{h}' = \mathbf{h} + \phi_s^{\text{token}}(\mathbf{h}^T)$, where $\phi_s^{\text{token}}(\cdot)$ operates on the transposed input to mix information across tokens; (2) **Channel Mixing**: The second block mixes information across channels (feature dimension) while keeping the token dimension constant: $\mathbf{h}'' = \mathbf{h}' + \phi_s^{\text{channel}}(\mathbf{h}')$, where $\phi_s^{\text{channel}}(\cdot)$ is a function to mix information across channels.

## E  DATASET INFORMATION

In this work, we selected several PDE systems from publicly available datasets (Kochkov et al., 2021; Rao et al., 2023): the 2D Burgers equation, the 2D and 3D Gray-Scott (GS) Reaction-Diffusion (RD) systems, a 2D real-world Black Sea (BS) dataset, and the 2D Unmanned Aerial Vehicle Maneuvering (UAVM) dataset. All data are sampled from high-resolution data with various space and time steps.

**2D Viscous Burgers Equation:** Generally, Burgers equation describes fluid dynamics with a simple non-linear convection–diffusion PDE on various input parameters. Within a given 2-dimensional (2D) field, a general evolution formulation of velocity $\mathbf{u} = [u,v]^T$ on per 2D grid point $\mathbf{x} = [x,y] \in \mathbb{R}^2$ at time $t \in \mathbb{R}^1$ is expressed as the following form:

$$\frac{\partial \mathbf{u}}{\partial t} + \mathbf{u} \cdot \nabla \mathbf{u} = \mathbf{D}\nabla^2\mathbf{u}, \tag{S13}$$

where viscosity $\mathbf{D} = [D_u, D_v]$ represents the diffusion coefficient of fluid.

For concreteness, all simulation trajectories are generated within $\Omega \in [0,1]^2$ and $t \in [0,1]s$, via a fourth-order Runge–Kutta (RK4) time integration method (Ren et al., 2022) under the periodic condition. Here, we define $D_u = D_v = 0.01, \Delta t = 0.001s$ and $\Delta x = 0.02$, referring to the parameters in (Ren et al., 2022; Mi & Sun, 2024).

**2D and 3D Gray-Scott (GS) Equations:** Generally, the Gray-Scott equations describe the chemical reaction between two substances $u$ and $v$, both of which diffuse over time. They also describe dynamical processes of non-chemical nature (e.g., physical dynamics). In this work, we define the form of Gray-Scott equation as a coupled reaction-diffusion PDE, consisting of velocity $\mathbf{u} = [u,v]^T$. Within a computational field, its corresponding form of each component in 2D space (e.g., $\mathbf{x} = [x,y] \in \mathbb{R}^2$) or 3D space (e.g., $\mathbf{x} = [x,y,z] \in \mathbb{R}^3$) and $t \in [0,T]$ is as follows:

$$\frac{\partial u}{\partial t} = D_u\nabla^2 u - uv^2 + \alpha(1-u), \tag{S14a}$$

$$\frac{\partial v}{\partial t} = D_v\nabla^2 v + uv^2 - (\beta + \alpha)v, \tag{S14b}$$

where $D_u$ and $D_v$ are the variable diffusion coefficients; $\beta$ is the conversion rate; $\alpha$ is the in-flow rate of $u$ from the outside; $(\alpha + \beta)$ is the removal rate of $v$ from the reaction field.

The 2D simulation trajectories are generated via the RK4 time integration method (Ren et al., 2022) within $\Omega \in [0,96]^2$ at $t \in [0,750]s$ and the 3D simulation trajectories are generated within $\Omega \in [0,48]^3$ at $t \in [0,750]s$ under the periodic condition. Here, we define $D_u = 0.2, D_v = 0.1, \alpha = 0.025, \beta = 0.055, \Delta t = 0.25s$ and $\Delta x = 2$, referring to the parameters in (Ren et al., 2022; Mi & Sun, 2024).

**2D Black Sea (BS) Dataset:** The Black Sea (BS) dataset is a real-world dataset, consisting of the realistic monitoring data within the Black Sea domain [1]. This dataset includes daily ocean fields of the BS basin, e.g., the averaged water flow velocities $u(x, y, t)$ and $v(x, y, t)$ and sea surface temperature $T(x, y, t)$, recorded from June 1, 1993 to June 30, 2021, with a horizontal resolution of $1/27° \times 1/36°$ and 31 levels of sea elevation.

In our work, we uniformly sample 1,000 points from approximately 40,000 real measurement points for training and inference. Note that we divide the data into annual units to effectively capture the periodic pattern in time series data, referring to the experimental setting in (Ren et al., 2022; Mi & Sun, 2024).

**2D Unmanned Aerial Vehicle Maneuvering (UAVM) dataset:** The UAVM dataset [2] provides complex airflow dynamics of an unmanned aerial vehicle maneuvering simulation within various environments. For example, there are three types of boundary interpolation functions (step function, linear function, and spline function) to be utilized for modeling turbulence with the software Ansys Fluent. The raw numerical simulation data is generated with the dynamical meshes and varying points (3,388 points in average after down-sampling process). In our paper, the velocity $u, v$ and the pressure $ps, pg$ are the input and output variables with 990 time-steps of 33 seconds for training and inference.

## F  BASELINE MODELS

To quantitatively evaluate the performance improvement of DSTGN, we selected the comprehensive set of baseline models, including graph-based models, transformer-based models, fno-based model, and some composite models. A summary of properties for all models is provided in Table S4.

**Graph Attention Network (GAT)** GAT (Velikovi et al., 2018) applied the self-attention mechanism into the graph network with the masked operation. The architecture of GAT consists of the encoder, multiple processor layers equipped with attention mechanisms, and a decoder. We list its parameters on various datasets in Appendix Section G, including the number of layers, the learning rate, the hidden dimension, etc. Relevant parameters are referenced from (Velikovi et al., 2018).

**Graph Attention Network Variant (GATv2)** GATv2 (Brody et al., 2022), a variant of GAT, defined a dynamic graph attention to solve the static attention issue in GAT. GATv2 removes the limitation of static attention of GAT in complex controlled problems. We list its parameters on various datasets in Appendix Section G, including the number of layers, the learning rate, the hidden dimension, etc. Relevant parameters are referenced from (Brody et al., 2022).

**MeshGraphNet (MGN)** MGN (Pfaff et al., 2021), a message passing neural network (Gilmer et al., 2017a), is well-suited for modeling physical systems represented as graphs. Its influential " Encoder-Processor-Decoder" architecture has been widely utilized in many tasks (e.g., the forward problem constrained by PDEs). We list the corresponding parameters on various datasets in Appendix Section G, including the number of layers, the learning rate, the hidden dimension, etc. Relevant parameters are referenced from (Gilmer et al., 2017a).

**MP-Neural-PDE Solver (MP-PDE):** MP-PDE solver (Brandstetter et al., 2022), a variant of MGN, proposed two primary training tricks (i.e., the temporal bundling and push-forward techniques) to alleviate the error accumulation in the autoregressive training strategy. A notable architectural difference is that MP-PDE solver adopts the convolution network as the decoder. We list the corresponding parameters on various datasets in Appendix Section G, including the number of layers, the learning rate, the hidden dimension, etc. Relevant parameters are referenced from (Brandstetter et al., 2022).

**Fourier Neural Operator (FNO):** FNO (Li et al., 2021), a promising spectral approach, attempts to learn the integral kernel operator $\kappa_t$ on the computational domain $\Omega$, described as the following form: $(\kappa_t(v_t))(x) = \int_\Omega k^{(t)}(x, y)v_t(y)dv_t(y)$, where $v_t$ is a Borel measure on $\Omega$. $x$ and $y$ is the spatial coordinates. $t$ is the time step. FNO constructs this operator in the Fourier domain to model the complex spatiotemporal dynamics. In our work, it was implemented for 2D and 3D spatial grid

---

[1]https://data.marine.copernicus.eu/product/BLKSEA_MULTIYEAR_PHY_007_004/description

[2]https://datasets.liris.cnrs.fr/eagle-version1

Table S4: Summary analysis of all models. The "Global Modeling" represents the approach that captures global interactions and dependencies, as opposed to localized processing of the "Local Modeling". The " Temporal Strategy" denotes the specific methodology employed for processing information across the time dimension. This defines how past, present, and future states are related and integrated. The "Spatial Strategy" signifies The specific methodology used for processing information across the spatial dimension. This defines how components interact and exchange information based on their spatial relationships or connectivity. The "Space Mapping" refers to the method employing the inter-spatial mapping between varying spaces (e.g., from irregular mesh to regular grid).

| Model | Regular Grid | Irregular Mesh | Global Modeling | Local Modeling | Temporal Strategy | Spatial Strategy | Space Mapping | Virtual Node |
|---|---|---|---|---|---|---|---|---|
| MLP | ✓ | ✓ | ✗ | ✗ | ✗ | ✗ | ✗ | ✗ |
| GAT | ✓ | ✓ | ✗ | ✓ | ✗ | ✓ | ✗ | ✗ |
| GATv2 | ✓ | ✓ | ✗ | ✓ | ✗ | ✓ | ✗ | ✗ |
| MGN | ✓ | ✓ | ✗ | ✓ | ✗ | ✓ | ✗ | ✗ |
| MP-PDE | ✓ | ✓ | ✗ | ✓ | ✓ | ✗ | ✗ | ✗ |
| FNO | ✓ | ✗ | ✓ | ✗ | ✗ | ✗ | ✗ | ✗ |
| FFNO | ✓ | ✗ | ✓ | ✗ | ✗ | ✓ | ✗ | ✗ |
| GeoFNO | ✓ | ✓ | ✓ | ✗ | ✗ | ✗ | ✓ | ✓ |
| Transformer | ✓ | ✓ | ✓ | ✗ | ✗ | ✗ | ✗ | ✗ |
| OFormer | ✓ | ✓ | ✓ | ✗ | ✗ | ✓ | ✗ | ✗ |
| Transolver | ✓ | ✓ | ✓ | ✗ | ✗ | ✓ | ✓ | ✓ |
| DSTGN | ✓ | ✓ | ✓ | ✓ | ✓ | ✓ | ✗ | ✗ |

domains. We list the corresponding parameters on various datasets in Appendix Section G, including the number of layers, the learning rate, the hidden dimension, etc. Relevant parameters are referenced from (Li et al., 2021).

**Factorized Fourier Neural Operator (FFNO):** Factorized Fourier Neural Operator (FFNO) (Tran et al., 2021), a variant of FNO, is an approach for simulating partial differential equations (PDEs). FFNO provide a new representations learning strategy with separable spectral layers, improving its scalability and outperforming the FNO model on several challenging benchmarks. We list its parameters on various datasets in Appendix Section G, including the number of layers, the learning rate, the hidden dimension, etc. Relevant parameters are referenced from (Tran et al., 2021).

**Geometry-informed FNO (GeoFNO):** Geometry-informed FNO (GeoFNO) (Li et al., 2024a), a variant of FNO, attempts to provide a mapping technique into neural operator by mapping the irregular domain to a uniform grid, adapting FNO model into handling the arbitrary geometries and preserving the computation efficiency. We list its parameters on various datasets in Appendix Section G, including the number of layers, the learning rate, the hidden dimension, etc. Relevant parameters are referenced from (Li et al., 2024a).

**Transformer:** Transformer is a foundational model (Vaswani et al., 2017) for natural language processing (NLP) tasks and is characterized by the use of self-attention mechanism: $\text{softmax}\left(QK^T/\sqrt{d_k}\right)V$, where $Q$ is the query matrix, $K$ is the key matrix, $V$ is the value matrix, $d_k$ is the dimension of the key vector (used for scaling). It is designed to process sequences of data by encoding input representations and decoding outputs without relying on recurrent or convolutional layers. We list its parameters on various datasets in Appendix Section G, including the number of layers, the learning rate, the hidden dimension, etc. Relevant parameters are referenced from (Vaswani et al., 2017).

**Operator Transformer (OFormer):** OFormer (Li et al., 2023) proposes an attention-based framework for data-driven operator learning, which is built upon the self-attention and cross-attention mechanisms under some assumption on the sampling pattern of input functions (or query locations). A similar work is the Galerkin Transformer (Cao, 2021a), which is the sub-work ("Galerkin" type) of OFormer (Li et al., 2023). We list its parameters on various datasets in Appendix Section G, including the number of layers, the learning rate, the hidden dimension, etc. Relevant parameters are referenced from (Li et al., 2023).

**Transolver:** Transolver (Wu et al., 2024), a variant of Transformer, attempts to capture the intricate correlations in physics with a physics-attention. By splitting the 2D or 3D space into the learnable

slices, Transolver also adopted the similar concept in GeoFNO to provide a generalization ability for a diversity of applications with the linear complexity. We list its parameters on various datasets in Appendix Section G, including the number of layers, the learning rate, the hidden dimension, etc. Relevant parameters are referenced from (Wu et al., 2024).

# G  TRAINING SETTINGS

**Overall settings:** All models are trained over 1,000 epochs. The encoder module include several functions (e.g., node encoder, edge encoder) with a 2-layer MLP with a hidden size of 128, and the decoder has a 2-layer MLP with a hidden size of 128. All the latent feature dimension is set as 128 and all the MLPs are the 2-layer architecture. We train models with the ReduceLROnPlateau Scheduler, Adam Optimizer, MSE loss function and evaluates by RMSE metric. Notably, all the hyperparameters are trained on various ranges. For example, the learning rate is selected from the set $\{10^{-1}, 10^{-2}, 10^{-3}, 10^{-4}, 10^{-5}\}$ and the batch size is selected from the set $\{1, 2, 5, 10, 20, 30, 40, 50, 100\}$.

Generally, the default setting is that the number of processor layers is set to 4, the standard deviation of noise is set to 0.0001, the number of attention head is set to 8, the learning rate and latent feature dimension are uniformly fixed at 0.0001 and 128 respectively, across all datasets. Moreover, we encode the cyclical timestamp features for handling the periodic patterns in the 2D BS dataset. All models adopt consistent settings across most benchmarks, with task-specific adjustments where necessary, described as follows.

**DSTGN:** The number of processor layers is increased to 10 for the UAVM dataset. The standard deviation of noise is set to 0.01 for the BS system and 0.02 for UAVM. The batch size is set to 30 for the Burgers and 2D GS systems, 2 for 3D GS, 20 for BS, and 1 for UAVM.

**GAT:** The number of processor layers is set to 10 for UAVM. The standard deviation of noise is configured as 0.01 for BS and 0.02 for UAVM. Batch sizes are set to 100 for the Burgers and 2D GS systems, 5 for 3D GS, 20 for BS, and 1 for UAVM.

**GATv2:** The number of processor layers is set to 4 for the Burgers, 2D GS, 3D GS, and BS systems, and 10 for UAVM. The noise standard deviation is set to 0.01 for BS and 0.02 for UAVM. Batch sizes are configured as 100 for the Burgers and 2D GS systems, 5 for 3D GS, 20 for BS, and 1 for UAVM.

**MGN:** The number of processor layers is set to 10 for UAVM. The noise standard deviation is configured as 0.01 for BS and 0.02 for UAVM. Batch sizes are set to 100 for the Burgers and 2D GS systems, 5 for 3D GS, 20 for BS, and 1 for UAVM.

**MP-PDE:** The number of processor layers is set to 10 for UAVM. The standard deviation of noise is set to 0.01 for BS and 0.02 for UAVM. The batch size is configured as 100 for the Burgers and 2D GS systems, 5 for 3D GS, 20 for BS, and 1 for UAVM. The rollout step is uniformly set to 2 across all benchmarks.

**FNO:** Batch sizes are set to 100 for the Burgers and 2D GS systems, and 10 for 3D GS. The number of Fourier modes is uniformly set to 8 for all benchmarks.

**FFNO:** Batch sizes are configured as 100 for the Burgers and 2D GS systems, and 10 for 3D GS. The number of Fourier modes is uniformly set to 8 across all benchmarks.

**GeoFNO:** The number of processor layers is set to 10 for UAVM. The noise standard deviation is set to 0.01 for BS and 0.02 for UAVM. Batch sizes are set to 100 for the Burgers and 2D GS systems, 10 for 3D GS, 20 for BS, and 1 for UAVM. The number of Fourier modes is uniformly set to 8. The grid size for geometric mapping is configured as $32\times32$ for the Burgers, 2D GS, and BS systems, $10\times10\times10$ for 3D GS, and $20\times20$ for UAVM.

**Transformer:** Transformer uses 6 processor layers for 3D GS, 10 for UAVM, and 8 for others. The noise standard deviation is set to 0.01 for BS and 0.02 for UAVM. Batch sizes are 30 for Burgers and 2D GS, 2 for 3D GS, 20 for BS, and 1 for UAVM.

**OFormer:** OFormer is configured with 10 processor layers for UAVM and 8 for others. The standard deviation of noise is set to 0.01 on BS and 0.02 on UAVM. Batch sizes are set to 100 for the Burgers and 2D GS systems, 10 for 3D GS, 20 for BS, and 1 for UAVM.

Table S5: Results on 2D Burgers with and without training noise.

| Case | DSTGN | Transolver | Transformer | MP-PDE | MGN |
|---|---|---|---|---|---|
| with noise | **0.0065** | **0.1742** | **0.0374** | **0.0178** | **0.0117** |
| **w/o** noise | 0.0069 | 0.1768 | 0.0388 | 0.0192 | 0.0129 |

Table S6: Results on 2D Burgers under similar parameter volume and inference time.

| Case | DSTGN | Transolver | Transformer | MP-PDE | MGN |
|---|---|---|---|---|---|
| Case 1 | **0.0065** | 0.1742 | 0.0374 | 0.0178 | 0.0117 |
| Case 2 | **0.0065** | 0.1741 | 0.0377 | 0.0321 | 0.0198 |

**Transolver:** Transolver uses 10 processor layers for UAVM and 8 for others. The noise standard deviation is set to 0.01 for BS and 0.02 for UAVM. Batch sizes are 100 for the Burgers and 2D GS systems, 10 for 3D GS, 20 for BS, and 1 for UAVM. The slice size is set to 32 across all benchmarks. The grid size for mapping is $8 \times 8$ for the Burgers, 2D GS, BS, and UAVM systems, and $8 \times 8 \times 8$ for the 3D GS system.

# H  ADDITIONAL RESULTS

## H.1  EFFECT OF INJECTING TRAINING NOISE

In our work, we consider a Gaussian noise injection strategy (Sanchez-Gonzalez et al., 2020; Pfaff et al., 2021) in training stage for stabilizing the model and enhancing the generalization ability for learning spatiotemporal dynamics. We conducted experiments of DSTGN and several representative baselines on 2D Burgers equation under fair conditions to explore the effect of injecting training noise. See Table S5 for the comparison of removing training noise and injecting training noise. The results demonstrate that the injection of training noise improves the inference performances of all models.

## H.2  COMPARISON UNDER OTHER SETTINGS

Given the above setups, we further conducted experiments of DSTGN and several representative baselines on 2D Burgers equation under other conditions, including the similar parameter volume (Case 1) and the similar inference time (Case 2). To further ensure the fairness of comparison, we increase the number of processor layer in some models (e.g., MGN and MP-PDE) for a similar parameter volume (roughly 2 million parameters). Likewise, a nearly operation is applied to ensure the consistence of inference time (0.008 s). See Table S6 for the comparison results. All the experimental results support the effectiveness of our method and its superiority over other baselines.

## H.3  RUNNING TIME AND MEMORY INFORMATION

Given the above setups, we provide the training time, the inference time and the memory overhead of DSTGN and several representative baselines on 2D Burgers equation in Table S7. Also, we provide an ablation study on 2D Burgers equation (see Table S7 below), showing the model's efficiency when the temporal module is removed. Notably, without the temporal module, our approach becomes more efficient while still achieving competitive results.

## H.4  HYPERPARAMETER TEST

Given the above setups, we provide the hyperparameter (e.g., hidden layer, Std. of noise, and $k$ value in S2 Processor) sensitivity study of DSTGN on 2D Burgers equation in Tables S8-S10. The results for node scaling test are also listed in Table S11.

Table S7: Running time and memory information.

| Model | DSTGN | DSTGN **w/o** T | Transolver | Transformer | MP-PDE | MGN |
|---|---|---|---|---|---|---|
| Training time (s) | 45.4 | 30.1 | **29.5** | 92.3 | 36.5 | 44.2 |
| Inference (s) | 0.0074 | **0.0062** | 0.0082 | 0.0070 | 0.0087 | 0.0079 |
| GPU usage (GB) | 28.1 | **19.0** | 19.9 | 56.6 | 65.8 | 64.1 |
| RMSE | **0.0065** | 0.0077 | 0.1741 | 0.0377 | 0.0321 | 0.0198 |

Table S8: Sensitivity study of hyperparameter: Layer.

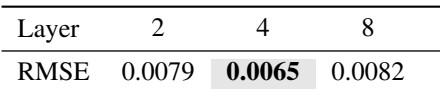

| Layer | 2 | 4 | 8 |
|---|---|---|---|
| RMSE | 0.0079 | **0.0065** | 0.0082 |

Table S9: Sensitivity study of hyperparameter: Std. of noise.

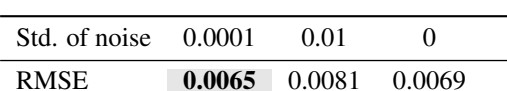

| Std. of noise | 0.0001 | 0.01 | 0 |
|---|---|---|---|
| RMSE | **0.0065** | 0.0081 | 0.0069 |

Table S10: Sensitivity study of hyperparameter: $k$ value in S2 Processor.

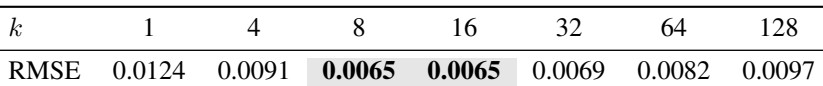

| $k$ | 1 | 4 | 8 | 16 | 32 | 64 | 128 |
|---|---|---|---|---|---|---|---|
| RMSE | 0.0124 | 0.0091 | **0.0065** | **0.0065** | 0.0069 | 0.0082 | 0.0097 |

Table S11: Node scaling test.

| Node number | 1,600 | 2,500 | 4,900 | 10,000 |
|---|---|---|---|---|
| RMSE | 0.0048 | 0.0065 | 0.0070 | 0.0093 |

Table S12: Results on other typical dynamic datasets.

| Dataset | Cylinder Flow | Flag Dataset |
|---|---|---|
| MGN | 0.1579 | 0.4038 |
| DSTGN | **0.0954** | **0.2158** |

## H.5 RESULTS ON THE TYPICAL DYNAMIC DATASETS

In this part, we further conducted experiments on the 2D cylinder flow (CF) (training/validation/test: 200/10/10) and the 3D Flag (100/10/10) datasets, and provide the results in Table S12 below. All results demonstrate a robust generalizability of our model (e.g., compared with MGN).

## H.6 IMPACT ON THE 1D EXAMPLE

In this part, we provide the results on the 1D Korteweg-de Vries (KdV) equation (training/validation/test: 5/5/5) in Table S13 below. The results of two test cases demonstrate the effectivness of T1 block.

Table S13: Results on 1D case.

| Model | 1D KdV |
|-------|--------|
| S1 | 0.6328 |
| S1 + T1 | 0.5804 |
| DSTGN | **0.5651** |

Table S14: Results of DSTGN with different $\Delta t$.

| $\Delta t$ | 0.001 s | 0.002 s | 0.003 s |
|-----------|---------|---------|---------|
| MGN | 0.0117 | 0.0125 | 0.0143 |
| DSTGN | **0.0065** | **0.0067** | **0.0074** |

Table S15: Results of DSTGN and multiscale models.

| Model | 2D Burgers |
|-------|-----------|
| MGN  (Pfaff et al., 2021) | 0.0117 |
| MsMGN  (Fortunato et al., 2022) | 0.0103 |
| AMGNet (Yang et al., 2022) | 0.0109 |
| HCMT (Yu et al., 2024) | 0.0296 |
| BSMS-GNN (Cao et al., 2023) | 0.0104 |
| UPT (Alkin et al., 2024) | 0.2009 |
| DSTGN | **0.0065** |

### H.7 IMPACT OF DIFFERENT $\Delta t$ ON DSTGN

In this part, we provide the results of DSTGN on 2D Burgers with different $\Delta t$ in Table S14 below.

### H.8 COMPARISON WITH MULTISCALE METHODS

In this part, we further conducted experiments on the 2D Burgers datasets, and provide the results of DSTGN and other multiscale methods in Table S15 below. We can see that our DSTGN achieved the best performance.

### H.9 EVOLUTION SNAPSHOTS OF DSTGN

In this part, we provide the snapshots of all modes on five benchmarks at some time points, shown in Figures S2 to S3. Moreover, we exhibit the evolution of DSTGN on five dataset, show in Figures S4 to S8.

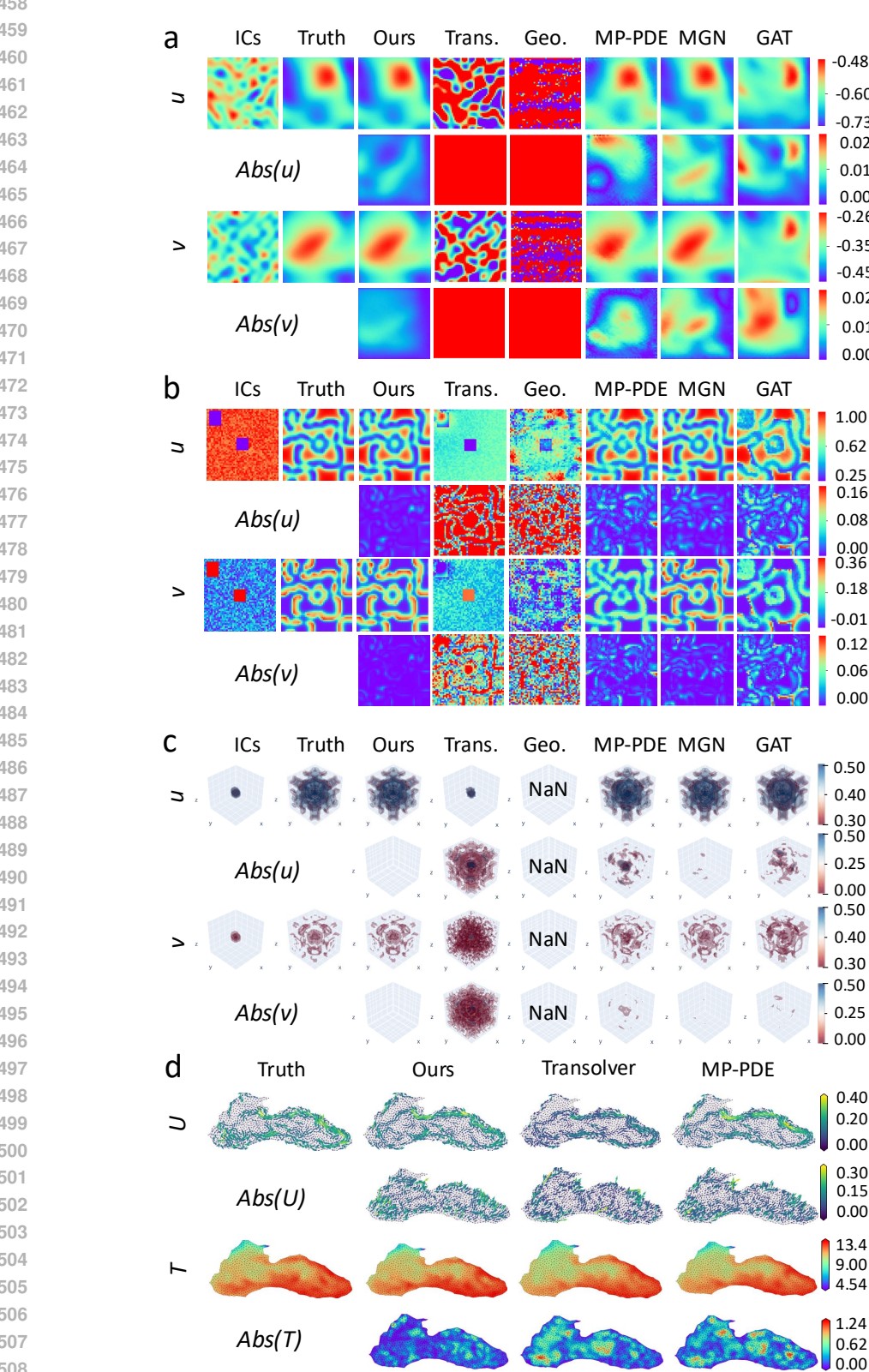

Figure S2: Visualization snapshots of UStMGN and various baselines on PDE systems and the BS dataset. **a-c**, the snapshots at the last time step on three PDE systems. **d**, the snapshots at the 4th time step on 2D BS dataset.

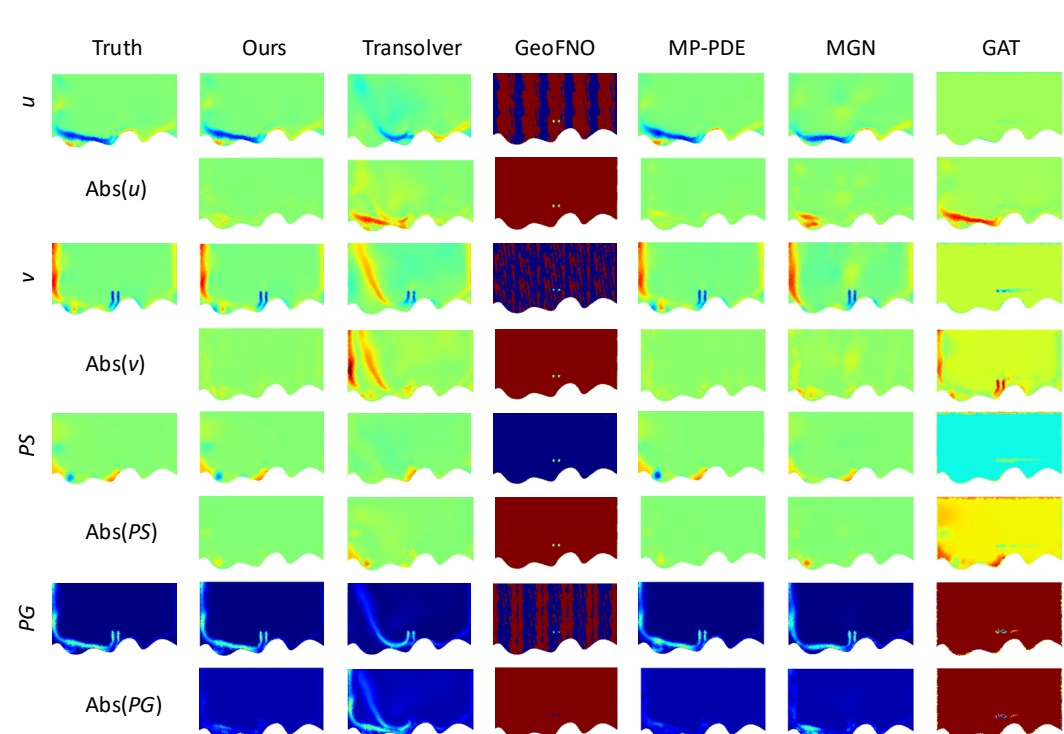

Figure S3: Visualization snapshots of DSTGN and various baselines on 2D UVAM dataset at $32s$.

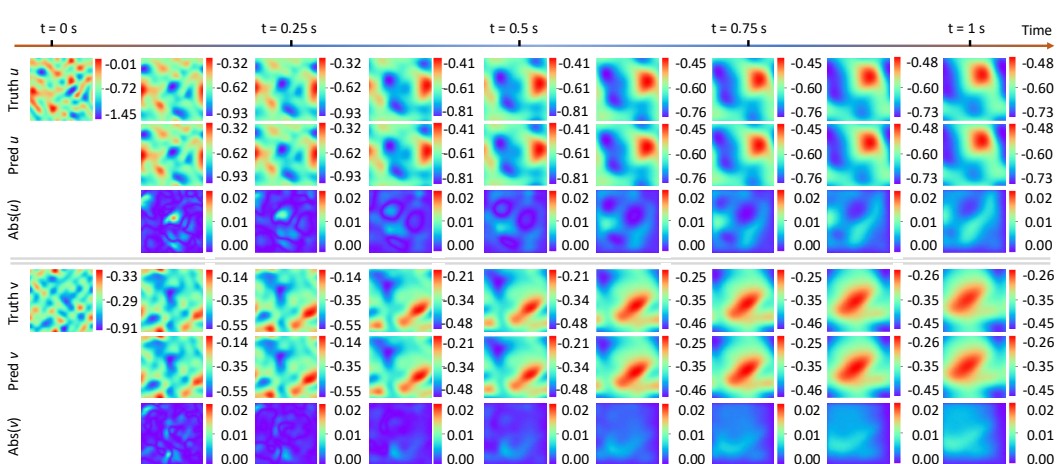

Figure S4: Visualization snapshots of DSTGN on 2D Burgers equation from 0 s to 1 s.

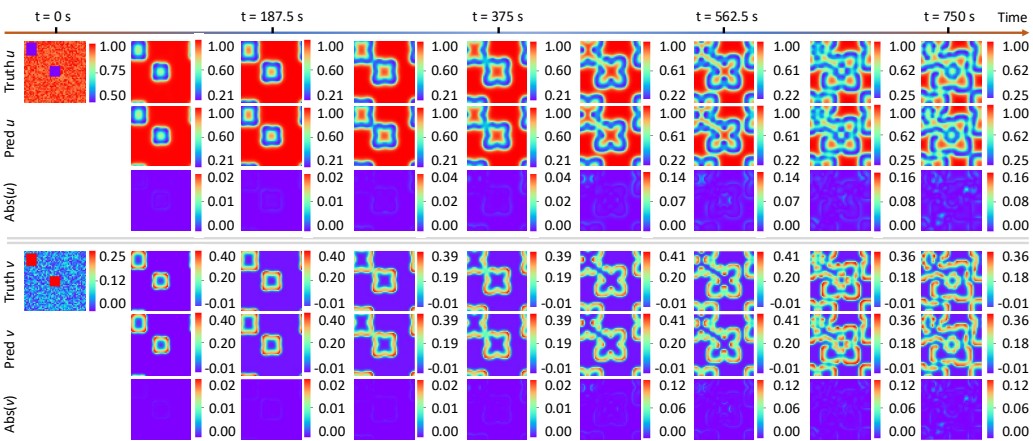

Figure S5: Visualization snapshots of DSTGN on 2D GS equation from 0 s to 750 s.

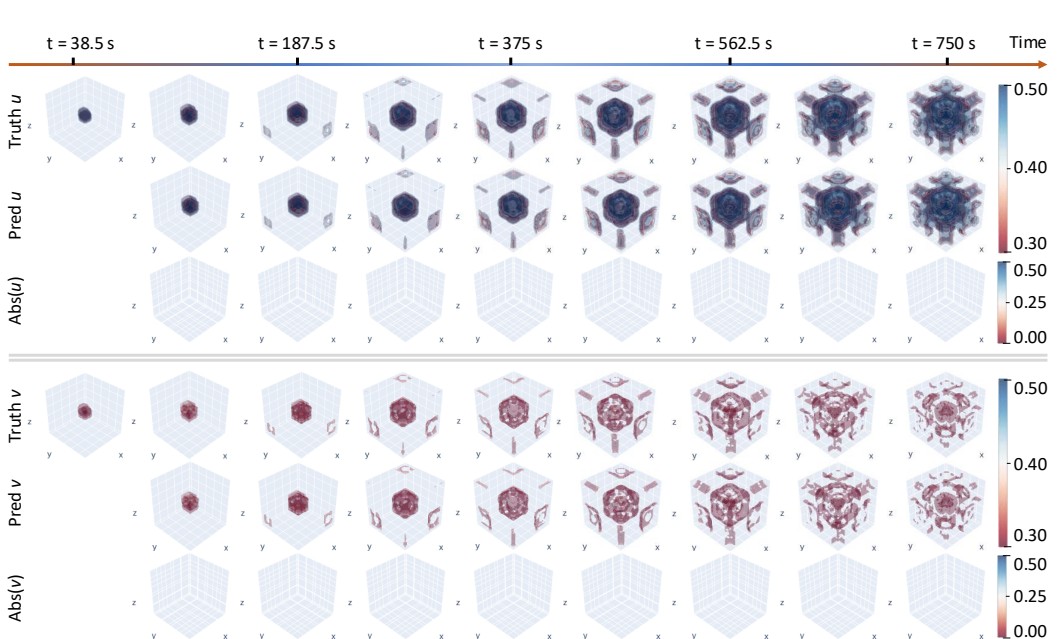

Figure S6: Visualization snapshots of DSTGN on 3D GS Dataset from 38.5 s to 750 s.

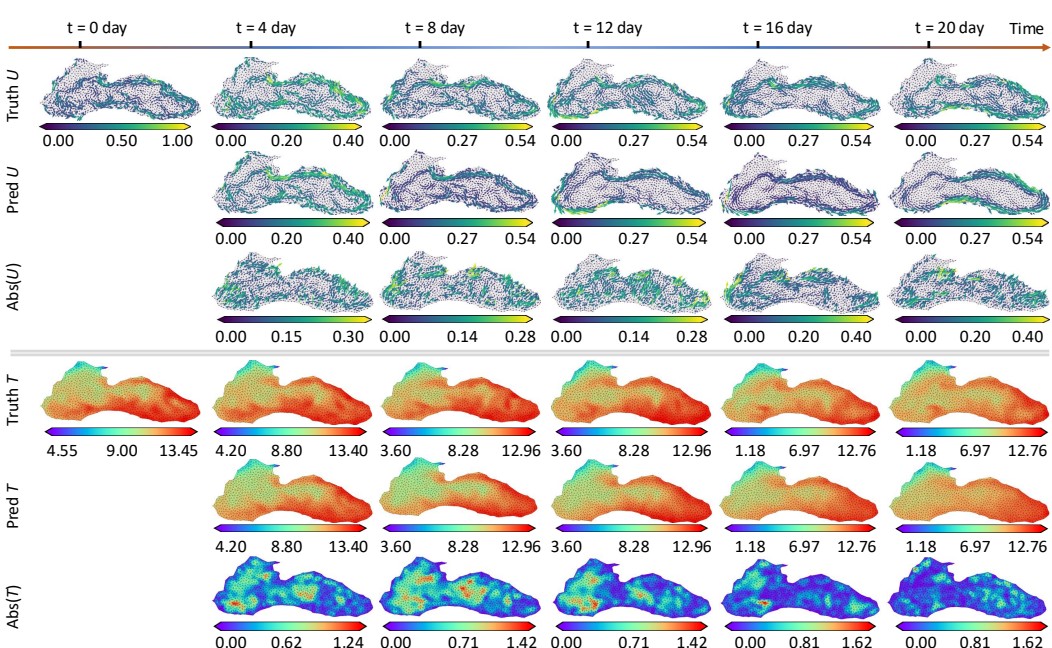

Figure S7: Visualization snapshots of DSTGN on 2D BS Dataset from 0 day to 20 day.

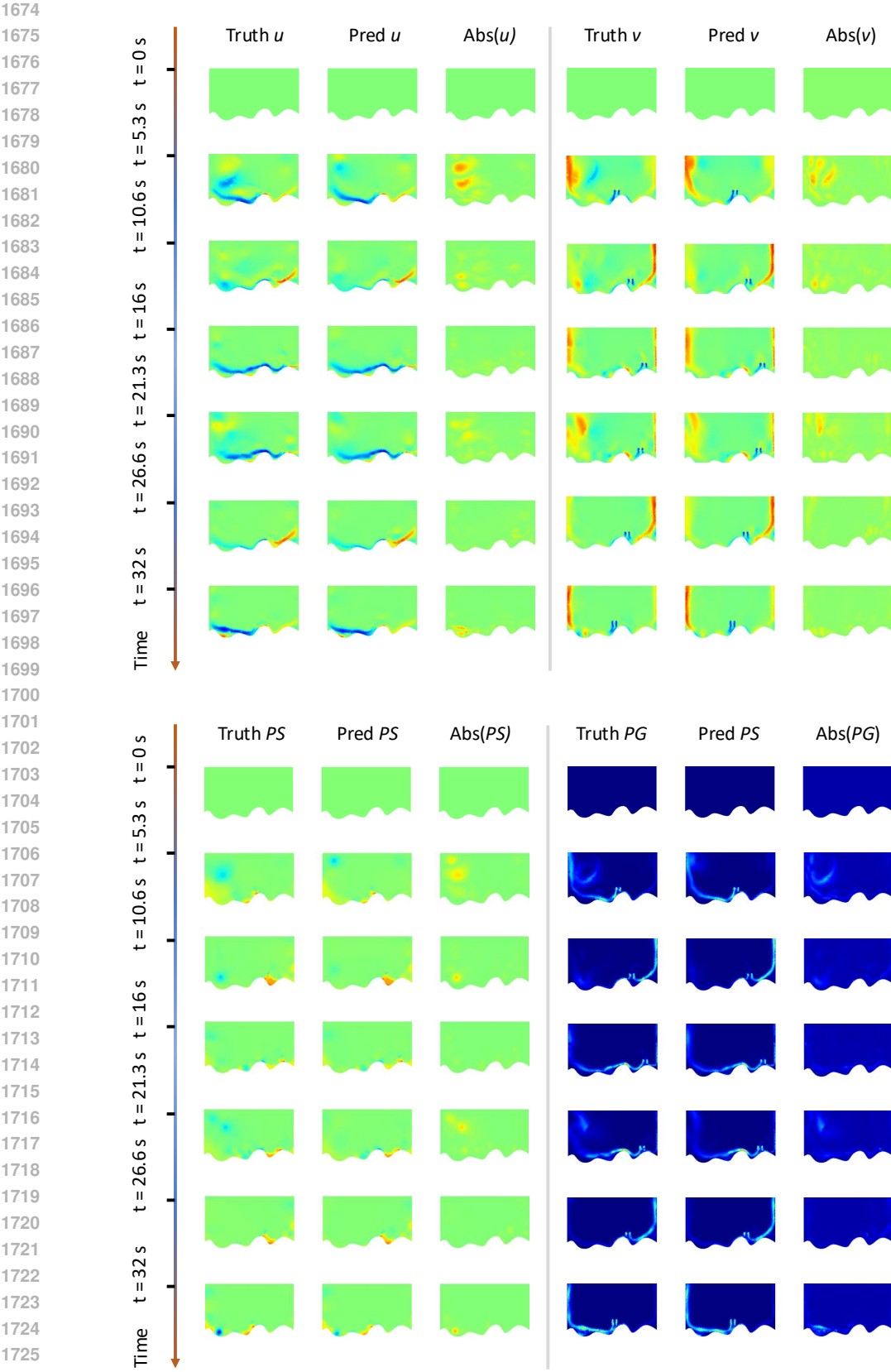

Figure S8: Visualization snapshots of DSTGN on 2D UAVM Dataset from 0 s to 33 s.

