# OpenReview forum: "DSTGN: Decoupled SpatioTemporal Graph Network for Multi-scenario Dynamics Learning"
_ICLR.cc/2026/Conference — ICLR 2026 Conference Withdrawn Submission_

### Official Review · Reviewer_DGQb · 2025-10-15

**Soundness:** 3
**Presentation:** 3
**Contribution:** 2
**Rating:** 4
**Confidence:** 4

**Summary:**

This paper presents Decoupled SpatioTemporal Graph Network (DSTGN), a framework that improves GNN-based dynamic prediction by addressing the limitations of local aggregation. DSTGN decouples spatial connectivity in both physical and latent spaces to better capture global interactions and introduces a learnable temporal integration mechanism to reduce error accumulation. Experiments show that DSTGN consistently outperforms existing methods in accuracy and generalization across diverse simulation domains.

**Strengths:**

- The paper is clearly written and well-structured, making it easy to follow the motivation, methodology, and results.
- The paper proposes a learnable temporal integration strategy, inspired by predictor–corrector schemes, introduces a novel and theoretically grounded perspective that distinguishes this work from standard message-passing approaches.
- The experimental evaluation is extensive, covering diverse settings including 2D/3D PDE systems, real-world oceanographic data, and dynamic UAV simulations. The inclusion of detailed ablation studies further strengthens the empirical validation of each architectural component.

**Weaknesses:**

- The decoupling in the physical space, which separates the graph into nodes and edges for message passing, is conceptually aligned with the design used in MeshGraphNets and most existing GNN-based simulators. As such, this component does not introduce a clear methodological novelty.
- The proposed S2 and T1 blocks essentially perform additional forms of message aggregation and transformation within the latent space and temporal domain. While they provide performance gains, the overall architecture still closely resembles MeshGraphNets with auxiliary feature fusion.
- The current predictor–corrector temporal strategy is restricted to a second-order formulation. Although effective for moderate systems, it may not remain stable under chaotic or stiff dynamics. Extending the framework to higher-order or adaptive integration schemes could further enhance long-term accuracy and robustness.

**Questions:**

1. In table 2, the S1 configuration appears to correspond directly to the MeshGraphNets baseline, while the variants (S1 + S2, S1 + T1, S2 + T1, and the full model) add additional blocks and, therefore, a larger number of parameters. Are the improvements due to design or model size? Please clarify whether parameter counts were controlled for fair comparison.
2. The term “Order” in the first column of Table 3 is not clearly defined. Please elaborate on what each “order” (e.g., S → T, T → S, S ∥ T) represents, how these learning patterns differ in the data flow or training process.

---

> ### Author Response · Authors · 2025-11-21
> **Rebuttal to the comments by Reviewer DGQb (Part 1)**
>
> We sincerely thank you for your comments. Below are our detailed responses to each point raised. We have also made comprehensive revisions (indicated in **red** color) and paragraph adjustments (indicated in **blue** color) of the paper, as shown in **the updated .pdf file**. Please see our **Global Reply** above for details.
>
> **Q1: The decoupling in the physical space, which separates the graph into nodes and edges for message passing, is conceptually aligned with the design used in MeshGraphNets and most existing GNN-based simulators. As such, this component does not introduce a clear methodological novelty.**
>
> **Reply:** Thanks for your comment! In fact, our model is built upon a decoupled spatio-temporal graph network, featuring dedicated modules for physical and latent-space modules (S1 & S2) alongside a temporal processor (T1). However, within the spatial modeling stage, the feature degradation issue in the message-passing mechanism (S1) may limit spatial learning, necessitating a new module to relieve it. Inspired by the clustering and diffusion of node information in physical spaces, we posit that high-dimensional latent features may exhibit analogous relational patterns. We thus generalize this phenomenon from the physical space to the latent space and accordingly propose a spatial learning block (S2) operating on latent representations. Note that the S1 is not a part of our contribution.
>
> For clarity, we have revised the contribution statement (see Lines 83-91 on Page 2) and would like to clarify the novelty of our paper as follows:
>
> - We propose a new spatial block that includes a channel-wise feature interaction learning module (S2) in the latent space as a supplement to standard node-wise message-passing-based GNNs, which, as a result, strengthens DSTGN's spatial learning capacity.
>
> - We regard the layer-wise feature update as a temporal marching process and thus establish a learnable time integration scheme (T1) to improve DSTGN's temporal modeling capacity, thereby reducing the autoregressive error accumulation. This design achieves a **learnable generalization** of the predictor–corrector scheme in the context of a neural network framework, which is architecturally distinct from existing post processing pattern of predictor–corrector method.
>
> Through these innovations, DSTGN offers a new framework for spatio-temporal learning, with demonstrated empirical advantages. In particular, DSTGN shows an superior long-term prediction performance for multi-scenario spatiotemporal learning across regular and irregular domains, as well as static and dynamic meshes.
>
> **Q2: The proposed S2 and T1 blocks essentially perform additional forms of message aggregation and transformation within the latent space and temporal domain. While they provide performance gains, the overall architecture still closely resembles MeshGraphNets with auxiliary feature fusion.**
>
> **Reply:** Thanks for your comment! In fact, our core contribution is a new architectural philosophy of decoupling, rather than simply aggregating features. Unlike MeshGraphNets and similar frameworks that employ tightly coupled spatio-temporal processing, our method explicitly decouples the process into spatial and temporal learning pathways (which aligns with the intrinsic spatio-temporally seperable nature of PDE systems). The S1, S2, and T1 blocks are the manifestations of this principle. This is not a minor augmentation but forms a principled framework. Our systematic decomposition from a coupled to a decoupled paradigm is the fundamental source of the performance gains, as it enables more focused and effective modeling of complex dynamics, which we demonstrate through comprehensive ablation studies.
>
> In addition, our model takes the following process: "*{node, edge} ⇒ Encoder (two MLPs for node, edge) ⇒ {S1 Processor (physical space), S2 Processor (latent space) ⇒ T1 Processor} $\times L$ ⇒ Decoder (MLP) ⇒ updated node features*". The full algorithm has been provided in Appendix Algorithm 1 (see Lines 1036-1059 on Page 20). Below is the detailed process of three blocks.
>
> - The S1 block is applied inthe physical space with the process of "*initialize node ($R^{n \times d}$),initialize edge ($R^{e \times d}$) ⇒ update the node and edge features via $\phi_{v}$ and $\phi_{e}$ ⇒ updated node ($R^{n \times d}$) and edge ($R^{e \times d}$*)".
>
> - The S2 block investigates the channel-level spatial interaction information in the latent spaces with the process of "*initialize node ($R^{n \times d}$) ⇒ update the features via a MLP ($R^{n \times k}$) ⇒ latent feature M ($R^{k \times d}$) ⇒ latent evolution ($R^{k \times d}$) with $\phi_{evo}$ (see Line 169 on Page 4) ⇒ updated node ($R^{n \times d}$)*".
>
> - The T1 block adopts the learnable predictor-corrector scheme, reducing autoregressive error accumulation.
>
> Therefore, our framework is fundamentally and conceptually distinct from MeshGraphNets with auxiliary feature fusion.

---

> > ### Author Response · Authors · 2025-11-21
> > **Rebuttal to the comments by Reviewer DGQb (Part 2)**
> >
> > **Q3: The current predictor–corrector temporal strategy is restricted to a second-order formulation. Although effective for moderate systems, it may not remain stable under chaotic or stiff dynamics. Extending the framework to higher-order or adaptive integration schemes could further enhance long-term accuracy and robustness.**
> >
> > **Reply:** Thanks for your comment! Our design is inspired by the Predictor-Corrector time integration scheme commonly seen in numerical methods. In fact, our T-block is a learnable generalization of such a scheme in the context of a neural network framework. In a strict sense, it is not restricted to a second-order formulation especially when the processor layer increases. Moreover, experimental results on **chaotic systems** (e.g., 2D and 3D reaction-diffusion systems) and **more challenging real-world dataset** (e.g., 2D Black Sea dataset) demonstrate our model's capablity. We plan to explore more temporal schemes on broader cases in our future evaluation.
> >
> > **Q4: In table 2, the S1 configuration appears to correspond directly to the MeshGraphNets baseline, while the variants (S1 + S2, S1 + T1, S2 + T1, and the full model) add additional blocks and, therefore, a larger number of parameters. Are the improvements due to design or model size? Please clarify whether parameter counts were controlled for fair comparison.**
> >
> > **Reply:** Thanks for your comment! Table 2 is an ablative study, whose **established common practice** is to evaluate the contribution of each component by adding or removing specific modules, without artificially controlling the number of parameters. The performance drop observed when removing a module (e.g., S1 in the S2+T1 configuration) directly demonstrates its importance.
> >
> > In fact, the performance gains in our full model stem primarily from our architectural design, not merely from increased parameters. For example, it is evidenced by the S2+T1 configuration, which uses additional parameters yet performs poorly, highlighting that the key factor is not parameter count but the effective components.
> >
> > For other cross-model comparisons, **fair and controlled conditions were ensured.** Models were constrained to a **similar parameter volume** (roughly 2 million parameters) and a **similar inference time** (0.008 s per inference step), which is described in Lines 457-461 on Page 9, guaranteeing a **fair and meaningful comparison**.
> >
> > **Q5: The term “Order” in the first column of Table 3 is not clearly defined. Please elaborate on what each “order” (e.g., S → T, T → S, S ∥ T) represents, how these learning patterns differ in the data flow or training process.**
> >
> > **Reply:** Thanks for your comment! In fact, Table 3 includes three **common modeling paradigms** prevalent in spatio-temporal forecasting. Due to the potential ambiguity of the term "order," we have substituted it with "Case" to avoid confusion.
> >
> > To further improve clarity, we have also revised the notation, defined as follows:
> >
> > - The symbol "S→T$^{\*}$" denotes a sequential architecture: "input features $h^{l}$ ⇒ S block (predictied states $h^{l+1,\*}$) ⇒ T block **only handling current states** (namely, exclusively processing states $h^{l+1,\*}$) ⇒ next updated features $h^{l+1}$".
> >
> > - Conversely, the symbol "T$^{\*}$→S" represents the reverse sequential order: "input features $h^{l}$ ⇒ T block **only handling current states** (namely, exclusively processing states $h^{l}$) ⇒ S block ⇒ updated states $h^{l+1}$.
> >
> > - The symbol "S||T$^{\*}$" signifies a parallel architecture: "input features $h^{l}$ ⇒ S block & T block (namely, concurrently processing states $h^{l}$) ⇒ next updated features $h^{l+1}$.
> >
> > We hope this clarifies your concern.
> >
> > **Concluding remark:** We sincerely thank you for constructive comments, which are greatly helpful in improving the clarity and quality of our paper. We look forward to addressing any additional questions. Your consideration of improving the rating of our paper will be much appreciated!

---

> ### Author Response · Authors · 2025-11-27
> **Looking forward to your feedback**
>
> Dear Reviewer DGQb,
>
> Thanks for your comments. We would like to follow up on our rebuttal to ensure that your concerns have been adequately addressed. If there are any further questions or points that need discussion, we will be happy to address them. Your feedback is invaluable in helping us improve our work, and we eagerly await your response.
>
> Thank you very much for your time and consideration.
>
> Best regards,
>
> The Authors

---

### Official Review · Reviewer_U4Pv · 2025-10-28

**Soundness:** 1
**Presentation:** 1
**Contribution:** 3
**Rating:** 2
**Confidence:** 3

**Summary:**

Summary: GNNs are not expressive in capturing global interactions. To mitigate this, the authors propose decoupled spatio-temporal GN to enhance prediction. Additionally, introduce learnable time integration to decouple inter-step dynamics. They introduce a predictor corrector framework for the time-stepper.

**Strengths:**

The authors propose a learnable time-integration strategy that leverages a predictor-corrector setup.
The authors have conducted rigorous experimentation to showcase the improved performance of their architecture

**Weaknesses:**

1. Several grammatical errors. Needs to be improved.
2. Section 3.1.1 jumps into the architecture directly, without providing any introduction. The virtual intermediate variable for instance is a term that is not defined prior to 3.1.1.
3. “ there is a unknown” – poor writing.
4. “exploiting more virtual intermediate variables leads to a denser distribution,” – not clear what this means.
5. “a intractable issue arises as the dimensionality increases” - dimensionality of what?
6. “Thus, we retain only the first two components in Eq. 1” – as opposed to what? What are the remaining components? Why are they not retained?

**Questions:**

1. Could you explain how the model facilitates global modeling? There does not seem to be anything in section 3.1 that indicates that there is some form of global information aggregation being performed here. I would recommend re-writing all of 3.1.
2. The following claims are made in Table S4 but many of the terms used in the table are not mentioned elsewhere in the paper - Global modeling, temporal strategy, spatial strategy (What are these referring to? )
3. Overall the writing is very unclear and it is hard to differentiate the difference between the proposed architecture and UPT, with the exception of predictor-corrector within the time-stepper module.

---

> ### Author Response · Authors · 2025-11-21
> **Rebuttal to the comments by Reviewer U4Pv (Part 1)**
>
> We sincerely thank you for your comments. Below are our detailed responses to each point raised. We have also made comprehensive revisions (indicated in **red** color) of the paper, as shown in **the updated .pdf file**. Please see our **Global Reply** above for details.
>
> **Q1: Fix Typos.**
>
> **Reply:** Thanks for your meticulous reading. We have thoroughly revised the manuscript and corrected all typographical and grammatical errors after careful proofreading.
>
> **Q2: Explainations for writting concerns that the reviewer raised.**
>
> **Reply:** Thanks for your feedback. Following your comments, we have made careful revisions to improve the writing and clarity, which includes re-writing the contributions, re-orgnizing and re-writing of the Method section, re-drawing Figure 1 to better reflect the workflow of our method, as well as improving the definition of symbols. We believe we have made this clearer in the revised version. Please see our detailed responses to your questions as follows.
>
> **Q2.1: "Section 3.1.1 jumps into the architecture directly, without providing any introduction."**
>
> **Reply:** We have revised the Method section (please see the updated Section 3.1 on Page 3). To enhance contextual coherence, we **changed** "To realize the better performance of dynamics prediction" to "To address the above mentioned spatiotemporal problem" and **added** the title "Decoupled Spatiotemporal Graph Learning (DSTGN)" placed prior to the "network overview" paragraph. In this way, we firstly provide the problem description of spatiotemporal dynamics prediction and subsequently introduce our model to address it.
>
> **Q2.2: "The virtual intermediate variable for instance is a term that is not defined prior to 3.1.1."**
>
> **Reply:** Thank for your comment. For clarity, we **changed** the terminology of "virtual intermediate variable" to "virtual topological elements", and revised the corresponding description in Section 3.2 on Page 4. The revision reads:
>
> > "This graph structure (initially defined by a set of nodes and edges) can be enriched by introducing virtual topological elements (e.g., adjacent triangles) ..."
>
> **Q2.3: "exploiting more virtual intermediate variables leads to a denser distribution" – not clear what this means.**
>
> **Reply:** Thank for your comment. In fact, we intend to convey that, as the dimensionality of spatial simplex increases (illustrated in Fig. 2a), the introduction of additional intermediate topological elements consequently leads to a denser spatial distribution. Please see our revised description in Section 3.2 on Page 4, which reads:
>
> > "... can be enriched by introducing virtual topological elements (e.g., adjacent triangles). This transformation allows the network to capture ..."
>
> **Q2.4: "an intractable issue arises as the dimensionality increases" - dimensionality of what?**
>
> **Reply:** Thank for your comment. The dimensionality refers to "the dimensionality of the simplex". Please see our revised description in Section 3.2 on Page 4, which reads:
>
> > "However, incorporating more higher-order terms leads to rapidly escalating computational complexity as the simplex dimension increases."
>
> **Q2.5: "Thus, we retain only the first two components in Eq. 1" – as opposed to what? What are the remaining components? Why are they not retained?**
>
> **Reply:** Thank for your comment. The "remaining components" refers to virtual topological elements such as adjacent triangles and potentia higher-order terms. For clarity, we added $\sum_{t \in T} \mathcal{F}(\cdot) \Delta \gamma_t$ into the Eq. 1 and revised the corresponding description in Lines 181-183 on Page 4, which reads:
>
> > "... where $T$ denotes the set of virtual topological elements (e.g., triangles), and the ellipsis ($\cdots$) indicates potential higher-order terms."
>
> In addition, Eq. 1 attempts to **approximate the unknown** relationship between the discrete representation and the continuous space. However, incorporating more higher-order terms leads to rapidly escalating computational complexity as the simplex dimension increases. A foundational principle, striking a practical trade-off between performance and computational cost, is precisely **why** our model is designed to utilize and retain **only these first two components** in Eq. 1.
>
> **Q3: Could you explain how the model facilitates global modeling? There does not seem to be anything in section 3.1 that indicates that there is some form of global information aggregation being performed here.**
>
> **Reply:** Thanks for your comment! The global modeling is performed by the function $\phi_a$ of the T1 processor, which is responsible for modeling the interactions between global pieces of information within the phase of constructing a temporal affine mapping ($R^{N \times N}$). This occurs specifically in the temporal strategy of Section 3.2, rather than in Section 3.1. Please see our description on Section 3.2 in Line 266 on Page 5.

---

> > ### Author Response · Authors · 2025-11-21
> > **Rebuttal to the comments by Reviewer U4Pv (Part 2)**
> >
> > **Q4: The following claims are made in Table S4 but many of the terms used in the table are not mentioned elsewhere in the paper - Global modeling, temporal strategy, spatial strategy (What are these referring to? ).**
> >
> > **Reply:** Thanks for your comment! Following your suggestion, we have updated a more detailed caption to Appendix Table S4 that explains the specific criteria and justification for each entries. Note that the entries in Appendix Table S4 are intended not as novel claims but as **a summary of common properties** characterizing different models.
> >
> > - Global Modeling: An approach that captures global interactions and dependencies, as opposed to localized processing (see Lines 1088-1089 on Page 22).
> >
> > - Temporal Strategy: The specific methodology employed for processing information across the time dimension. This defines how past, present, and future states are related and integrated (see Lines 1090-1091 on Page 22).
> >
> > - Spatial Strategy: The specific methodology used for processing information across the spatial dimension. This defines how components interact and exchange information based on their spatial relationships or connectivity (see Lines 1092-1094 on Page 22).
> >
> > Correspondingly, in our work, the "Global Modeling" occurs in the temporal affine mapping ($R^{N \times N}$) constructed within the T1 processor, which captures global dynamics. The "Temporal Strategy" describes our dedicated T-block, which performs decoupled, step-wise temporal processing, built upon the predictor-corrector scheme. And the "Spatial Strategy" encompasses our dual spatial framework, consisting of the S1 (physical-space) and S2 (latent-space) modules.
> >
> > **Q5: Overall the writing is very unclear and it is hard to differentiate the difference between the proposed architecture and UPT, with the exception of predictor-corrector within the time-stepper module.**
> >
> > **Reply:** Thanks for your comment! Following your comments, we have made careful revisions to improve the writing and clarity, which includes re-writing the contributions, re-orgnizing and re-writing of the Method section, re-drawing Figure 1 to better reflect the workflow of our method, as well as improving the definition of symbols. We believe we have made this clearer in the revised version.
> >
> > In addition, we would like to clarify a fundamental distinction between our work (DSTGN) and UPT. The two architectures are substantially different in their core constructs: our model is built upon a decoupled spatio-temporal graph network, featuring dedicated modules for physical and latent-space graphs (S1 & S2) alongside a temporal processor (T1). In contrast, UPT employs a Transformer-based paradigm. Their detailed architecture are as follows:
> >
> > - DSTGN takes the following process: "*{node, edge} ⇒ Encoder (two MLPs for node, edge) ⇒ {S1 Processor (physical space), S2 Processor (latent space) ⇒ T1 Processor} $\times L$ ⇒ Decoder (MLP) ⇒ updated node features*", whose loss function for training the model only consists of data loss. The full algorithm has been provided in Appendix Algorithm 1 (see Lines 1036-1059 on Page 20). Here, the S1 block is a learning block in the physical spaces with the process of "*initialize node ($R^{n \times d}$),initialize edge ($R^{e \times d}$) ⇒ update the node and edge features via $\phi_{v}$ and $\phi_{e}$ ⇒ updated node ($R^{n \times d}$) and edge ($R^{e \times d}$*)". The S2 block is a learning block in the latent spaces with the process of "*initialize node ($R^{n \times d}$) ⇒ update the features via a MLP ($R^{n \times k}$) ⇒ latent feature M ($R^{k \times d}$) ⇒ latent evolution ($R^{k \times d}$) with $\phi_{evo}$ (see Line 169 on Page 4) ⇒ updated node ($R^{n \times d}$)*". The T1 block adopts the learnable predictor-corrector scheme, reducing autoregressive error accumulation.
> >
> > - UPT takes the following process: "*{node, edge} ⇒ GNN ⇒ Pooling (n supernodes) ⇒ Encoder (transformer) ⇒ Embedding new k latent vectors ⇒ latent perceiver (cross attention) ⇒ latent approximator (transformer) ⇒ Decoder (cross attention for unpooling) ⇒ updated node features*", whose loss function for training the model consists of data loss, reconstruction loss, and perceptual loss. In short, UPT adapts the "encoder-decoder" architecture with a Koopman-like operator scheme. This type of method typically requires the training data to be rich, relatively stationary, and not highly nonlinear.
> >
> > In summary, these two models (DSTGN & UPT) fundamentally differ from each other in terms of their motivation, design, core mechanics and the actual training strategy, placing them in separate categories. We hope this clarifies your concern.
> >
> > **Concluding remark:** We sincerely thank you for constructive comments, which are greatly helpful in improving the clarity and quality of our paper. We look forward to addressing any additional questions. Your consideration of improving the rating of our paper will be much appreciated!

---

> ### Author Response · Authors · 2025-11-27
> **Looking forward to your feedback**
>
> Dear Reviewer U4Pv,
>
> Thanks for your comments. We would like to follow up on our rebuttal to ensure that your concerns have been adequately addressed. If there are any further questions or points that need discussion, we will be happy to address them. Your feedback is invaluable in helping us improve our work, and we eagerly await your response.
>
> Thank you very much for your time and consideration.
>
> Best regards,
>
> The Authors

---

### Official Review · Reviewer_JggA · 2025-10-30

**Soundness:** 3
**Presentation:** 2
**Contribution:** 3
**Rating:** 6
**Confidence:** 3

**Summary:**

This paper focuses on learning dynamic systems and proposes a decoupled method that process spatial features and temporal update steps separately. The spatial encoder adapts an edge-based GNN and the temporal block includes the predictor-corrector sampling into the temporal update. The decoupled design enables the proposed DSTGN to achieve more precise temporal sampling process and mitigate the error accumulation issue.

**Strengths:**

1. It is a novel and valuable idea to introduce predictor-corrector updates for learning dynamic systems.
2. The decouple of spatial and temporal representations provides insights for future PDE-based models.
3. The experimental results validate the generalizability of DSTGN on a variety of dynamic systems.

**Weaknesses:**

1. While the spatial block of the proposed DSTGN is based on the message passing over node and edge embeddings, it does not fully exploit the high-order simplex as the authors have mentioned.
2. Most of the chosen baseline methods are neural operator models. Since the learning objective and architecture of DSTGN are more similar to mesh-based models, including more recent and relevant baselines should be considered to make a comprehensive comparison.
3. The learnable PC update module is proposed as a solution to error accumulation. However, its effectiveness is not theoretically guaranteed.

**Questions:**

1. Have you investigated any theoretical properties (e.g., stability, convergence) of the proposed temporal update rule?

---

> ### Author Response · Authors · 2025-11-21
> **Rebuttal to the comments by Reviewer JggA**
>
> We sincerely thank you for your comments. Below are our detailed responses to each point raised. We have also made comprehensive revisions (indicated in **red** color) and paragraph adjustments (indicated in **blue** color) of the paper, as shown in **the updated .pdf file**. Please see our **Global Reply** above for details.
>
> **Q1: While the spatial block of the proposed DSTGN is based on the message passing over node and edge embeddings, it does not fully exploit the high-order simplex as the authors have mentioned.**
>
> **Reply:** Thanks for your comment! We agree that higher-order schemes hold promise for capturing more complex dynamics, but they often entail significantly higher computational costs (as noted in Lines 165–170 on Page 4). Therefore, we intentionally retain only two components (i.e., node and edge) to balance performance and computational cost. Extending the framework to incorporate higher-order simplex structures remains an important direction for future work, as we continue to pursue high accuracy without prohibitive computational overhead.
>
> **Q2: Most of the chosen baseline methods are neural operator models. Since the learning objective and architecture of DSTGN are more similar to mesh-based models, including more recent and relevant baselines should be considered to make a comprehensive comparison.**
>
> **Reply:** Thanks for your comment! In our current work, we have included a variety of baseline models in Table 1, not only neural operators (e.g., FNO, FFNO, GeoFNO), but also graph- and mesh-based methods (e.g., GAT, GATv2, MeshGraphNet (MGN), Message Passing PDE Solver (MP-PDE)) and Transformer-based approaches (e.g., OFormer, Transolver). This design was to demonstrate the broad applicability and competitiveness of our DSTGN across different architectural families.
>
> In fact, we have also provided the experimental results of additional baseline models in Appendix Table S5 such as MsMGN [1], AMGNet [2], HCMT [3], BSMS-GNN [4], UPT [5]. We plan to incorporate more baseline models on broader cases in our future evaluation.
>
> **Q3: The learnable PC update module is proposed as a solution to error accumulation. However, its effectiveness is not theoretically guaranteed. Have you investigated any theoretical properties (e.g., stability, convergence) of the proposed temporal update rule?**
>
> **Reply:** Thanks for your comment! In this work, our temporal module is **inspired by** the Predictor-Corrector time integration scheme commonly seen in numerical methods. Specifically, our T-block is a learnable **generalization** of such a scheme in the context of a neural network framework.
>
> The stability and convergence of the classical Predictor-Corrector scheme (a semi-implicit time integration method) have been well-studied [6], which serves as theoretical basis for our proposed temporal scheme (see Section 3.2 and Appendix Section C). In fact, we have provided **the local truncation error analysis** of the predictor-corrector scheme as a theoretical support in Appendix Section C.3.
>
> Furthermore, we have demonstrated its effectiveness through **extensive empirical validation**, including the ablation study (see Tables 2-4), along with the final performance comparison (see Table 1). Its superior performance on various challenging systems serves as a strong evidence of its capability.
>
> **Concluding remark:** We sincerely thank you for constructive comments, which are greatly helpful in improving the clarity and quality of our paper. We look forward to addressing any additional questions!
>
> **References**
>
> [1] Fortunato M, Pfaff T, Wirnsberger P, et al. Multiscale meshgraphnets. Arxiv, 2022
>
> [2] Zhishuang Y, Yidao D, Xiaogang D, et al. Amgnet: multi-scale graph neural networks for flow field prediction. Connection Science, 34(1):2500–2519, 2022.
>
> [3] Youn-Yeol Y, Jeongwhan C, Woojin C, et al. Learning flexible body collision
> dynamics with hierarchical contact mesh transformer. ICLR, 2024.
>
> [4] Yadi C, Menglei C, Minchen L, et al. Efficient learning of mesh-based physical
> simulation with bsms-gnn. ICLR, 2024.
>
> [5] Benedikt A, Andreas F, Simon S, et al. Universal physics transformers: A framework for efficiently scaling neural operators. NIPS, 37:25152–25194, 2024.
>
> [6] Gragg W B, Stetter H J. Generalized multistep predictor-corrector methods[J]. Journal of the ACM (JACM), 11(2): 188-209, 1964.

---

> ### Author Response · Authors · 2025-11-27
> **Looking forward to your feedback**
>
> Dear Reviewer JggA,
>
> Thanks for your comments. We would like to follow up on our rebuttal to ensure that your concerns have been adequately addressed. If there are any further questions or points that need discussion, we will be happy to address them. Your feedback is invaluable in helping us improve our work, and we eagerly await your response.
>
> Thank you very much for your time and consideration.
>
> Best regards,
>
> The Authors

---

### Official Review · Reviewer_n9sA · 2025-10-31

**Soundness:** 3
**Presentation:** 3
**Contribution:** 2
**Rating:** 4
**Confidence:** 3

**Summary:**

This paper introduces a general framework named Decoupled SpatioTemporal Graph Network (DSTGN) for learning multi-scenario dynamics. DSTGN decouples spatial connectivity in both physical and latent spaces through S-blocks and introduces a learnable time integration mechanism via T-blocks to mitigate error accumulation in autoregressive inference. Experiments on multiple datasets demonstrate superior performance and generalization across static/dynamic and regular/irregular meshes.

**Strengths:**

- The paper is well-written and easy to follow.
- The spatial blocks consider both physical space and latent space interaction.
- Extensive experiments demonstrate the superior performance of the proposed method over the baselines.

**Weaknesses:**

- My major concern is novelty. Although the paper introduces the integration of decoupled spatial and temporal learning, most architectural components (e.g., encoder–processor–decoder, message passing, predictor–corrector) are adapted from existing works.
- The paper would benefit from a clear comparison of parameter counts across different models to better demonstrate computational efficiency and fairness in evaluation.
- In Table 3, several symbols and abbreviations are insufficiently explained, making it difficult for readers to fully understand.

**Questions:**

- Line 205: The paper mentions “channel cluster rather than node cluster.” Could the authors clarify what this means in practice and how the grouping operation is implemented?
- In Table 2, there is a notable performance drop when using the S2 + T1 configuration. Could the authors provide an explanation for why this combination leads to substantially weaker results?
- In Table 7, the reported running time of DSTGN appears faster than some baselines such as MGN. Could the authors elaborate on the factors contributing to this efficiency advantage?

---

> ### Author Response · Authors · 2025-11-21
> **Rebuttal to the comments by Reviewer n9sA (Part 1)**
>
> We sincerely thank you for your comments. Below are our detailed responses to each point raised. We have also made comprehensive revisions of the paper (indicated in **red** color), as shown in **the updated .pdf file**. Please see our **Global Reply** above for details.
>
> **Q1: My major concern is novelty. Although the paper introduces the integration of decoupled spatial and temporal learning, most architectural components (e.g., encoder–processor–decoder, message passing, predictor–corrector) are adapted from existing works.**
>
> **Reply:** Thank you for your feedback. We would like to clarify the novelty of our work along with the contributions in the following aspects:
>
> - We propose a new spatial block that includes a channel-wise feature interaction learning module (S2) in the latent space as a supplement to standard node-wise message-passing-based GNNs, which, as a result, strengthens DSTGN's spatial learning capacity.
>
> - We regard the layer-wise feature update as a temporal marching process and thus establish a learnable time integration scheme (T1) to improve DSTGN's temporal modeling capacity, thereby reducing the autoregressive error accumulation. This design achieves a learnable generalization of the predictor–corrector scheme in the context of a neural network framework, which is architecturally distinct from existing post processing pattern of predictor–corrector method.
>
> With these innovations in our model, DSTGN offers a new framework for spatio-temporal learning, with demonstrated empirical advantages. In particular, DSTGN shows an superior long-term prediction performance for multi-scenario spatiotemporal learning across regular and irregular domains, as well as static and dynamic meshes.
>
> These clarifications have been incorporated in the revised paper (please see Lines 83-91 on Page 2).
>
> **Q2: In Table 3, several symbols and abbreviations are insufficiently explained, making it difficult for readers to fully understand.**
>
> **Reply:** Thanks for your careful reading! Table 3 includes three **common modeling paradigms** prevalent in spatio-temporal forecasting. To improve clarity, we have revised the notation in the caption of Table 3, defined as follows:
>
> - The symbol "S→T$^{\*}$" denotes a sequential architecture: "input features $h^{l}$ ⇒ S block (predictied states $h^{l+1,\*}$) ⇒ T block **only handling current states** (namely, exclusively processing states $h^{l+1,\*}$) ⇒ next updated features $h^{l+1}$".
>
> - Conversely, the symbol "T$^{\*}$→S" represents the reverse sequential order: "input features $h^{l}$ ⇒ T block **only handling current states** (namely, exclusively processing states $h^{l}$) ⇒ S block ⇒ updated states $h^{l+1}$.
>
> - The symbol "S||T$^{\*}$" signifies a parallel architecture: "input features $h^{l}$ ⇒ S block & T block (namely, concurrently processing states $h^{l}$) ⇒ next updated features $h^{l+1}$.
>
> We hope this clarifies your concern.
>
> **Q3: Line 205: “channel cluster rather than node cluster.” Could the authors clarify what this means in practice and how the grouping operation is implemented?**
>
> **Reply:** Thanks for your question. Here is our response:
>
> **Meaning in Practice:** A "node cluster" groups nodes themselves, where each node belongs to a single cluster, focusing on **physical spatial locality**. In contrast, a "channel cluster" groups the **different feature channels of each node**. This allows a explicit feature channel interaction, focusing on **latent channel-specific locality**.
>
> - A simple case: For "node cluster", we usually groups 1,000 nodes into 10 clusters. For "channel cluster", we cluster nodes' 1024-dimensional feature tensor into 8 distinct groups.
>
> **Implementation:** In fact, the actual grouping process has been provided in Lines 204-213 on Page 4. In short, the S2 block takes the process of "*initialize node ($R^{n \times d}$) ⇒ update the features via a MLP ($R^{n \times k}$) ⇒ latent feature M ($R^{k \times d}$) ⇒ latent evolution ($R^{k \times d}$) with $\phi_{evo}$ ⇒ updated node ($R^{n \times d}$)*".
>
> **Q4: In Table 2, there is a notable performance drop when using the S2 + T1 configuration. Could the authors provide an explanation for why this combination leads to substantially weaker results?**
>
> **Reply:** Excellent remark! The performance drop with the S2+T1 configuration is, in fact, an expected outcome that validates our core design. Since DSTGN is fundamentally a graph model, its primary backbone is the graph-based module (i.e., S1). The weaker performance of S2+T1 in the ablation study precisely underscores the essential role of S1 in capturing spatial dependencies. Moreover, the complementary value of S2 and T1 is evident in other configurations in Table 2: both S1+S2 and S1+T1 outperform S1 alone, confirming that each module contributes unique capabilities to the overall framework.

---

> ### Author Response · Authors · 2025-11-21
> **Rebuttal to the comments by Reviewer n9sA (Part 2)**
>
> **Q5: In Table 7, the reported running time of DSTGN appears faster than some baselines such as MGN. Could the authors elaborate on the factors contributing to this efficiency advantage? The paper would benefit from a clear comparison of parameter counts across different models to better demonstrate computational efficiency and fairness in evaluation.**
>
> **Reply:** Thanks for your comment! In fact, the experiments in Table 7 were conducted under **strictly controlled conditions** (see Lines 457-461 on Page 9), including the similar parameter volume (roughly 2 million parameters) and the similar inference time (0.008 s), to **ensure the fairness of comparison**.
>
> Under such a case, the efficiency of DSTGN stems from its decoupled spatio-temporal architecture, which enables more robust performance compared to tightly coupled baselines like MGN.
>
> **Concluding remark:** We sincerely thank you for constructive comments, which are greatly helpful in improving the clarity and quality of our paper. We look forward to addressing any additional questions. Your consideration of improving the rating of our paper will be much appreciated!

---

> ### Author Response · Authors · 2025-11-27
> **Looking forward to your feedback**
>
> Dear Reviewer n9sA,
>
> Thanks for your comments. We would like to follow up on our rebuttal to ensure that your concerns have been adequately addressed. If there are any further questions or points that need discussion, we will be happy to address them. Your feedback is invaluable in helping us improve our work, and we eagerly await your response.
>
> Thank you very much for your time and consideration.
>
> Best regards,
>
> The Authors

---

### Author Response · Authors · 2025-11-21
**Global Reply: Remarks on the revised paper**

Dear Reviewers,

We wish to express our sincere gratitude for your time and for the constructive suggestions provided on our manuscript. Your feedback has been invaluable in helping us improve the quality and clarity of our work. We have carefully considered all the points raised and have revised the manuscript accordingly, including comprehensive revisions (indicated in **red** color) and paragraph adjustments (indicated in **blue** color) in **the updated .pdf file**. A update summary is provided below:

- **Clarification of novelty and contributions**: We have revised the descriptions of our contributions (see Lines 83-88 on Page 2) to enhance clarity and impact. This update has been consistently reflected in the "Abstract" (see Lines 18-19 on Page 1), "Introduction" (see Lines 76-80 on Page 2), and "Conclusion" (see Lines 476-478 on Page 9) for coherence.

- **Reorganization of Section "Methodology"**: We have restructured the Methodology section from a **sequential** description to a **general-to-specific** presentation. Concretely, we have added a title "Decoupled Spatiotemporal Graph Learning (DSTGN)" placed prior to the "network overview" paragraph. Accordingly, the "encoder" paragraph and the "decoder" paragraph now follow the "network overview" paragraph, rather than being embedded within the S-block and the T-block. Additionally, the title "Spatial learning in physical spaces" has been removed to prevent the potential misinterpretation that the S1 processor constitutes our primary contribution.

- **Enhanced Clarity in Figure and Notation**: We have redesigned Figure 1 (see Lines 54-69 on Page 2) to more accurately illustrate the method's workflow. The definitions of key symbols in Table 3 (see Lines 446-448 on Page 9) and Appendix Table S4 (see Lines 1188-1194 on Page 23) have also been expanded and standardized to ensure notational consistency throughout the paper.

Once again, we are deeply thankful for your thorough review. We hope that our revisions provided herein have fully addressed all your concerns.

Best rewards,

The Authors.

---

### Note · Authors · 2026-01-28

I have read and agree with the venue's withdrawal policy on behalf of myself and my co-authors.

---

### Meta-Review · Area_Chair_m2B6 · 2026-01-06

**Summary:**

The reviewers raised several concerns, the main ones are:

1. Novelty (as raised by reviewer n9sA and DGQb), where most architectural components (e.g., encoder–processor–decoder, message passing, predictor–corrector) are adapted from existing works. This is the main concern.

2. Lack of relevant baselines, especially mesh-based ones (by reviewer JggA).

3. The learnable PC update module is proposed as a solution to error accumulation. However, its effectiveness is not theoretically guaranteed (by reviewer JggA).

4. Clarity of writing (by reviewer U4Pv)

**Reviewer Concerns:**

The concerns 2, 3 and 4 are mostly addressed. However, I think the main concern 1 about novelty is not addressed. Specifically, I agree with reviewers n9sA and DGQb that most architectural components are adapted from existing works and the proposed method lacks enough novelty compared to prior methods (such as MGN, etc.) for a poster presentation.

**Reviewer Scores:**

After the rebuttal, I think reviewer U4Pv shall raise the score from 2 to 4. Other reviewers will most likely remain their scores.

---

### Decision · Program_Chairs · 2026-01-26

Reject